# Dynamic tuneable G protein-coupled receptor monomer-dimer populations

Patricia M. Dijkman[1,5], Oliver K. Castell[2,3], Alan D. Goddard[1,6], Juan C. Munoz-Garcia[1,7], Chris de Graaf [4], Mark I. Wallace [2,8] & Anthony Watts[1]

G protein-coupled receptors (GPCRs) are the largest class of membrane receptors, playing a key role in the regulation of processes as varied as neurotransmission and immune response. Evidence for GPCR oligomerisation has been accumulating that challenges the idea that GPCRs function solely as monomeric receptors; however, GPCR oligomerisation remains controversial primarily due to the difficulties in comparing evidence from very different types of structural and dynamic data. Using a combination of single-molecule and ensemble FRET, double electron–electron resonance spectroscopy, and simulations, we show that dimerisation of the GPCR neurotensin receptor 1 is regulated by receptor density and is dynamically tuneable over the physiological range. We propose a "rolling dimer" interface model in which multiple dimer conformations co-exist and interconvert. These findings unite previous seemingly conflicting observations, provide a compelling mechanism for regulating receptor signalling, and act as a guide for future physiological studies.

[1] Biomembrane Structure Unit, Department of Biochemistry, University of Oxford, South Parks Road, Oxford OX1 3QU, UK. [2] Department of Chemistry, University of Oxford, South Parks Road, Oxford OX1 3QU, UK. [3] School of Pharmacy and Pharmaceutical Sciences, College of Biomedical and Life Sciences, Cardiff University, King Edward VII Avenue, Cardiff CF10 3NB, UK. [4] Division of Medicinal Chemistry, Faculty of Sciences, Amsterdam Institute for Molecules, Medicines and Systems (AIMMS), Vrije Universiteit Amsterdam, De Boelelaan 1108, 1081 HZ Amsterdam, The Netherlands. [5] Present address: Max Planck Institute for Biophysics, Max-von-Laue-Straße 3, 60438 Frankfurt am Main, Germany. [6] Present address: School of Life and Health Sciences, Aston University, Aston Triangle, Birmingham B4 7ET, UK. [7] Present address: School of Pharmacy, University of East Anglia, Norwich Research Park, Norwich NR4 7TJ, UK. [8] Present address: Department of Chemistry, King's College London, Britannia House, 7 Trinity Street, London SE1 1DB, UK. These authors contributed equally: Patricia M. Dijkman, Oliver K. Castell. Correspondence and requests for materials should be addressed to M.I.W. (email: mark.wallace@kcl.ac.uk) or to A.W. (email: anthony.watts@bioch.ox.ac.uk)

G protein-coupled receptors (GPCRs) are the largest class of cell surface receptors forming the target of ~40% of marketed pharmaceuticals[1]. Over the past few decades the idea that GPCRs function as isolated monomeric receptors has been challenged by the accumulation of evidence for oligomerisation[2]. It is now widely accepted that constitutive dimerisation is essential for class C GPCR activity[3], and using a variety of techniques, an array of class A receptors has also been observed to oligomerise (reviewed by Ferré et al.[2]). Whilst monomeric GPCRs can efficiently couple to G proteins and recruit arrestin in vitro[4,5], this does not preclude the existence of functional oligomers in vivo. Indeed, receptor oligomerisation may be required to traffic receptors to the plasma membrane[6], and regulate receptor internalisation[7], but may also affect ligand binding[8], and G protein activation[9] suggesting some role for dimerisation in biased signalling[10]. Observations of cooperative ligand-binding

have strengthened the case for functional oligomerisation[11], although this interpretation has been challenged by others[12].

The controversies surrounding GPCR oligomerisation arise from apparent inconsistencies in both structural information and dynamics[2]. Class A GPCRs lack the large extramembranous domains that stabilise class C GPCR dimers[3], and oligomerisation has been proposed to occur primarily through transmembrane (TM) domain interactions[12]. Given the structural homology of class A GPCR TM domains, a common dimerisation mechanism might be envisaged. However, detailed morphological data for class A GPCR dimers remain largely elusive and are often conflicting[2], with every receptor transmembrane segment having been implicated in dimerisation and even studies on the same receptor proposing different interfaces (Supplementary Table 1).

Although many ensemble methods, including Förster resonance energy transfer (FRET), cross-linking, and

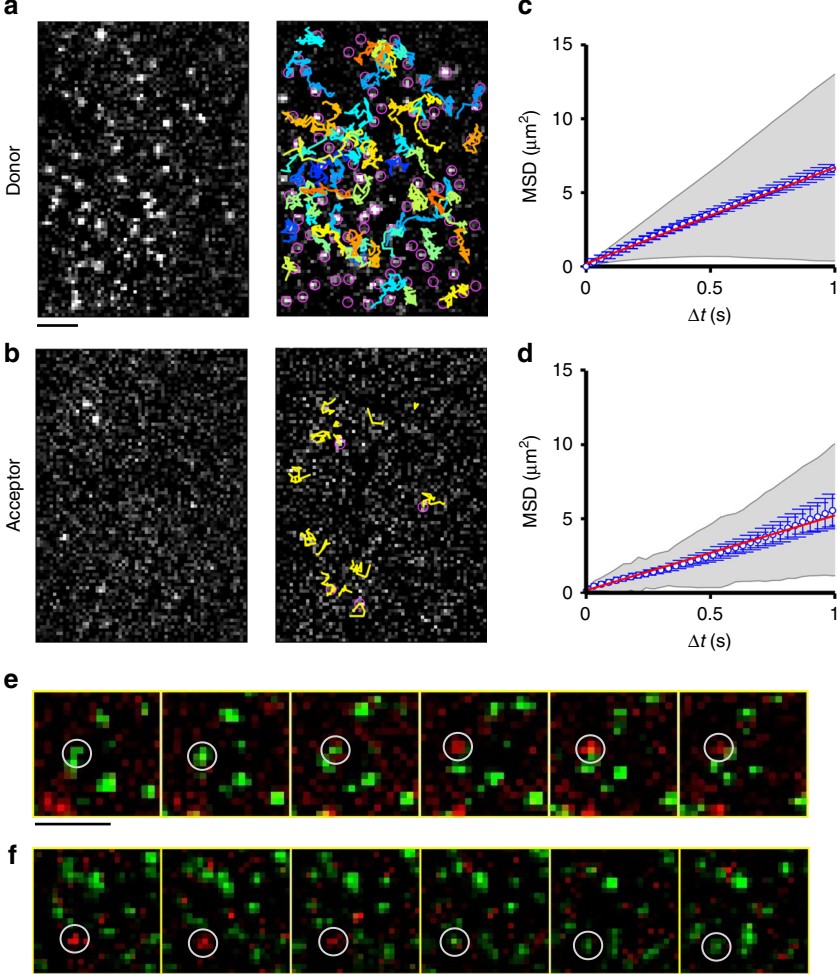

**Fig. 1** Single-molecule FRET trajectories. FRET **a** donor and **b** acceptor single-molecule fluorescence was imaged on a single camera. Single particles were detected and tracked in time. Images shown here following image processing (see Methods section). In the case of donor trajectories only a sub-fraction are overlaid for clarity. Scale bar: 5 µm. Two-dimensional diffusion constants were determined for the **c** donor- and **d** acceptor-imaged particles: $D_{donor}$ (monomer) = 1.63 µm² s⁻¹ (95% CI [1.609, 1.647] µm² s⁻¹, $n$ = 17,853 trajectories); $D_{acceptor}$ (dimer) = 1.27 µm² s⁻¹ (95% CI [1.222,1.317] µm² s⁻¹, $n$ = 1167 trajectories). Data points represent weighted average at each delay interval over all trajectories. Grey shaded area represents the weighted standard deviation over all individual trajectory mean squared displacement (MSD) curves. Error bars represent standard error of the mean on each data point. **e**, **f** Sequential frames (30 ms per frame) of overlaid donor (green—Cy3) and acceptor (red—Cy5) channels show partial trajectories of monomeric and dimeric NTS1 species. Scale bar: 5 µm. **e** Dimer formation: an initial population of Cy3-labelled monomeric NTS1 and dimeric Cy3- and Cy5-labelled NTS1 diffuse within the membrane. Between frames 3 and 4 a single Cy3-labelled receptor (encircled) switches from green to red indicative of the initiation of FRET as a result of dimer formation. See also Supplementary Movie 2. **f** Dimer termination of observation: a FRET capable NTS1 dimer comprised of Cy3- and Cy5-labelled receptor (encircled) is observed to diffuse in the membrane. Between frames 3 and 4 this spot switches from red to green as FRET is terminated, either as a result of receptor dissociation or acceptor photobleaching. See also Supplementary Movie 3

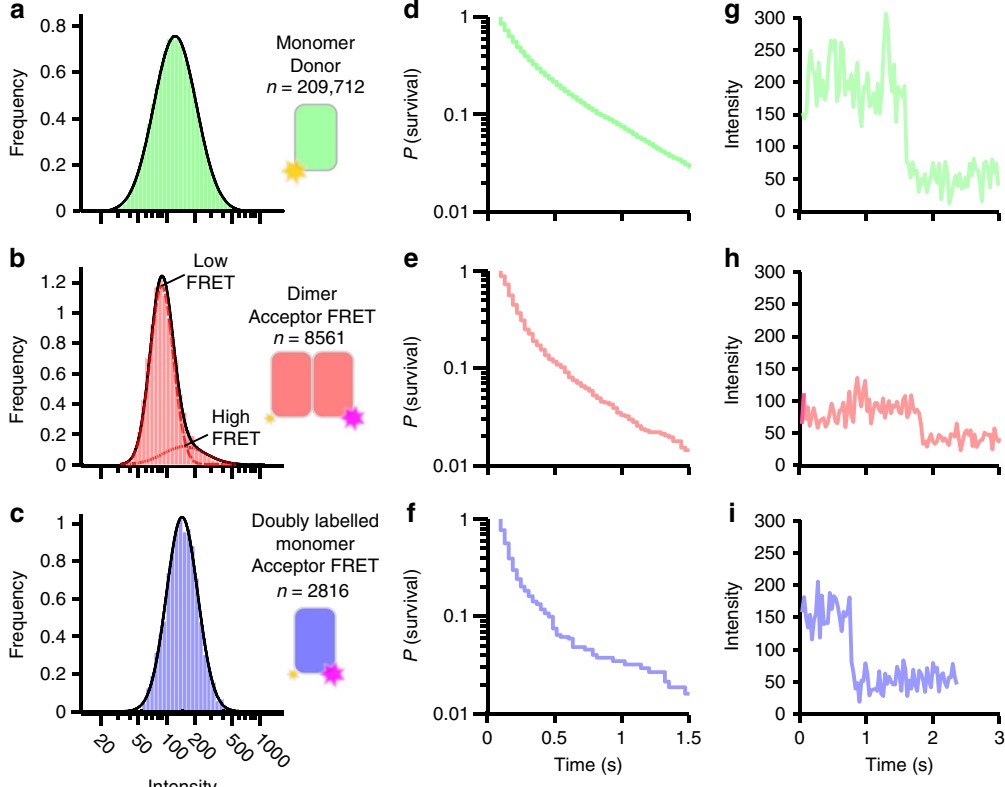

**Fig. 2** Single-molecule FRET intensity distributions. The distribution of single particle spot intensities is described well by a single log-normal distribution for **a** the predominantly monomeric donor spots ($n = 209,712$), and **c** for acceptor spots from a monomeric doubly labelled (TM1–TM4, Cy3–Cy5 distance ~1.5 nm) NTS1 control sample ($n = 2816$), while for **b** the FRET acceptor dimer spots it was described by the sum of two log-normal distributions representing high (dotted line) and low (dashed line) FRET dimeric conformations ($n = 8561$). Spot lifetime (**d–f**) was similar for all populations indicating dimer lifetime on the order of or greater than the photobleaching timescales. Single-step photobleaching (**g–i**) precludes higher-order oligomers. See also Supplementary Figs. 6, 7, and 8

co-immunoprecipitation have suggested GPCRs form constitutive oligomers[2], more recent single-molecule imaging techniques have revealed transient interactions. Cell-based FRAP studies showed homodimers of $D_2$ dopamine[13] and $\beta_1$-adrenergic receptors[14] were transient on the second timescale, but at expression levels likely to be well above native. Single-molecule imaging at lower expression levels of the M1 muscarinic acetylcholine receptor[15], $N$-formyl peptide receptor[16] and β1- and β2-adrenergic receptors[17,18] also showed transient dimerisation. However, even in these experiments, membrane organisation beyond intrinsic receptor–receptor interactions below the diffraction limit cannot be ruled out. Although these experiments provide good evidence for a dynamic equilibrium between GPCR monomer and dimer populations in vivo, the complexities of the cellular environment make it very difficult to dissect these isolated observations and construct a quantitative model of how and why this occurs.

The class A GPCR neurotensin receptor 1 (NTS1) has been implicated in schizophrenia and Parkinson's, has been postulated as a biomarker for various cancers[19], and has recently been linked to obesity[20]. Its peptide ligand, neurotensin (NT), has a dual role; it functions as a neurotransmitter in the central nervous system and as a local hormone in the intestines. NTS1 is also one of a small number of GPCRs that can be expressed recombinantly in *Escherichia coli*[21]. NTS1 therefore possesses the attractive combination of being both pharmaceutically important and a tractable model GPCR. Co-immunoprecipitation of NTS1 demonstrated homodimers in HeLa cells[22], but biophysical characterisation of NTS1 oligomerisation has been limited. At high receptor densities (~$10^3$ μm$^{-2}$), NTS1 is predominantly

dimeric (~90%) in liposomes[21]. In solution at low detergent concentrations, NTS1 dimerisation occurs in a concentration-dependent manner and dimers show ligand-binding cooperativity, but catalyse G protein nucleotide exchange with lower affinity than monomers[23].

Here, we combine single molecule and ensemble FRET with double electron–electron resonance (DEER) spectroscopy and simulations to study NTS1 dimerisation. Our findings support a "rolling interface" model for transient dimerisation in a concentration-dependent manner that is tuneable over the physiological range. These findings are in line with recent observations for other GPCRs, and rationalise previously incompatible observations and provide a putative mechanism for regulation of receptor signalling by dimerisation in vivo, extending the traditional ligand-dependent view of receptor modulation.

## Results

**Dynamics of NTS1 dimerisation.** To gain insight into dynamics of NTS1 dimerisation, single-molecule FRET (smFRET) experiments were conducted on receptors fluorescently labelled at the intracellular end of TM4 (T186C$^{4.42}$, where the superscript refers to Ballesteros-Weinstein numbering for GPCRs, see also Supplementary Fig. 1), which has previously been proposed to sit at the NTS1 dimerisation interface[24]. The cysteine depleted background mutant was shown to bind neurotensin with similar affinity as wild-type NTS1 (Supplementary Fig. 2), and T186C$^{4.42}$ labelled with Alexa Fluor 488 (A488) was confirmed to be capable of interacting with $Gα_{i1}$ in a GTPγS-dependent manner

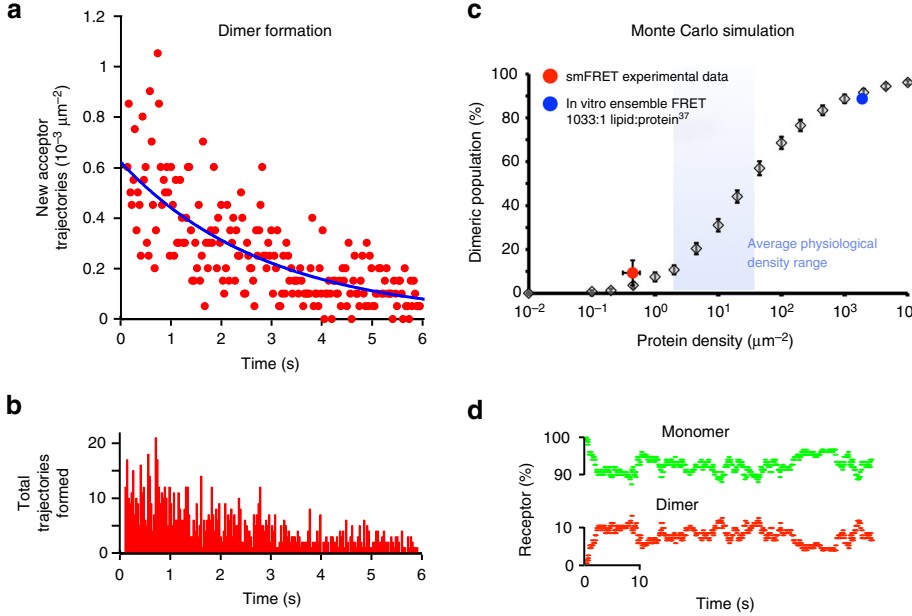

**Fig. 3** Dynamics of dimer population. **a** Experimental data: rate of Cy3–Cy5 dimer formation as observed by the formation of new acceptor trajectories ($n = 19$ videos). The observed frequency of dimer formation decreases with time due to photobleaching. Exponential fitted model (blue line) enables extrapolation to zero photobleaching and subsequent calculation of total dimer formation rate. **b** Experimental absolute frequency of observed dimer formation events. **c** The dimeric population observed during Monte Carlo simulations at different protein density parameterised by measured single-molecule receptor dynamics is shown. Simulations at the receptor density used in single-molecule experiments correlate well with experimental observations (red dot), and ensemble FRET measurements from previous studies at four orders of magnitude higher receptor density (blue dot). Blue shaded area highlights the reported physiological range of average cell surface receptor densities for NTS1 (2–40 μm$^{-2}$)[45]. **d** Monte Carlo simulation of single-molecule experimental conditions of receptor density and number of receptors (see also Supplementary Movie 4). Dynamic equilibrium illustrates fluctuation of monomer and dimer populations over the experimental timescale, explaining the observed variance in dimer formation (**a**, **b**) and dynamic equilibrium monomer-dimer distribution (**c**—red spot error bars, see also Supplementary Fig. 4)

(Supplementary Fig. 3), indicating that the purified receptor is functional. Based on the geometry of NTS1[25,26], the range of inter-label distances in the NTS1 dimer is expected to fall within 1.5–8 nm regardless of dimer configuration. The Cy3-Cy5 FRET pair used here has a Förster radius in the middle of this distance range ($R_0 = 5.4$ nm). Labelled receptor was reconstituted into DPhPC droplet interface bilayers[27] (Supplementary Fig. 1) at a 1:4 donor-to-acceptor ratio (see Methods section) and at low receptor densities (0.43 μm$^{-2}$) to mimic physiological conditions. Fluorescence emission from the donor-labelled (Cy3) and acceptor-labelled (Cy5) NTS1 was imaged using total internal reflection fluorescence (TIRF) microscopy with a temporal resolution of 30 ms. Diffraction-limited diffusive spots were observed in both the donor and acceptor channels with differing densities, corresponding to a majority population of monomeric protein and a smaller sub-population of FRET capable putative dimers (Fig. 1a, b and Supplementary Movie 1). In superimposed donor and acceptor videos, individual spots could be observed to transition from donor to acceptor and acceptor to donor (Fig. 1e, f) consistent with dimer formation and dissociation or acceptor photobleaching, respectively.

Two-dimensional diffusion constants were determined from the mean squared displacement of single trajectories in both the donor and acceptor channel (Fig. 1c, d). NTS1 molecules observed in the acceptor channel (i.e., dimers, $D_{acceptor} = 1.27$ μm$^2$ s$^{-1}$, 95% confidence interval (CI) [1.222,1.317] μm$^2$ s$^{-1}$, $n = 1167$ trajectories) diffused significantly more slowly than those in the donor channel (i.e., monomers, $D_{donor} = 1.63$ μm$^2$ s$^{-1}$, 95% CI [1.609,1.647] μm$^2$ s$^{-1}$, $n = 17,853$ trajectories) as gauged by a two-sided $z$-test ($z = -14.12$, $p < 10^{-5}$). Using the Saffman-Delbrück model for lateral diffusion ($D \propto -\ln[r]$), this is suggestive of an 1.4-fold increase in the effective particle radius ($r$),

supporting the assignment of acceptor spots as dimers[28]. The spot intensity distribution in both the donor and acceptor channel broadly fitted a lognormal distribution[29] with single-step photobleaching (Fig. 2), characteristic of single molecules.

The relative number of donor and acceptor spots provides a snapshot of the equilibrium monomer-dimer distribution (Supplementary Fig. 4). The frequency of observation of dimeric species fell sharply with decreasing receptor density (Supplementary Fig. 4a). After correcting for labelling efficiency (see Supplementary Note 1) we find that, on average, 9.8% of all receptors exist as dimers at a receptor density of 0.43 molecules per μm$^2$. False positive dimerisation detection due to co-diffusing, non-interacting monomers was quantified to be negligible (0.04%, Supplementary Note 2, and Supplementary Fig. 5). The frequency of observed dimerisation events decreased with time due to photobleaching of the donor population (Fig. 3a, b). The dimerisation rate constant was determined by extrapolation of the observed frequency of dimer formation to $t = 0$ s giving the rate of dimer formation in the absence of photobleaching (Fig. 3a and Supplementary Note 1). Following correction for labelling ratio we calculate a dimerisation $k_{on}$ of 0.081 μm$^{-2}$ s$^{-1}$ (95% CI [0.072,0.088] μm$^{-2}$ s$^{-1}$, Supplementary Note 1–4). Spot lifetime (Fig. 2d–f) was similar for all populations, indicating dimer lifetime largely on the order of, or greater than, the photobleaching timescales. Whilst this precluded accurate direct measurement of dimer lifetime, this could be calculated using the $k_{on}$ and the measured equilibrium between monomer and dimers (i.e. 9.8% dimers), giving a dimer half-life of $t_{1/2} = 1.2$ s, (95% CI [0.65,1.54] s).

**Concentration-dependence of dimerisation.** Due to the inherent limitations of single-molecule and ensemble techniques, an

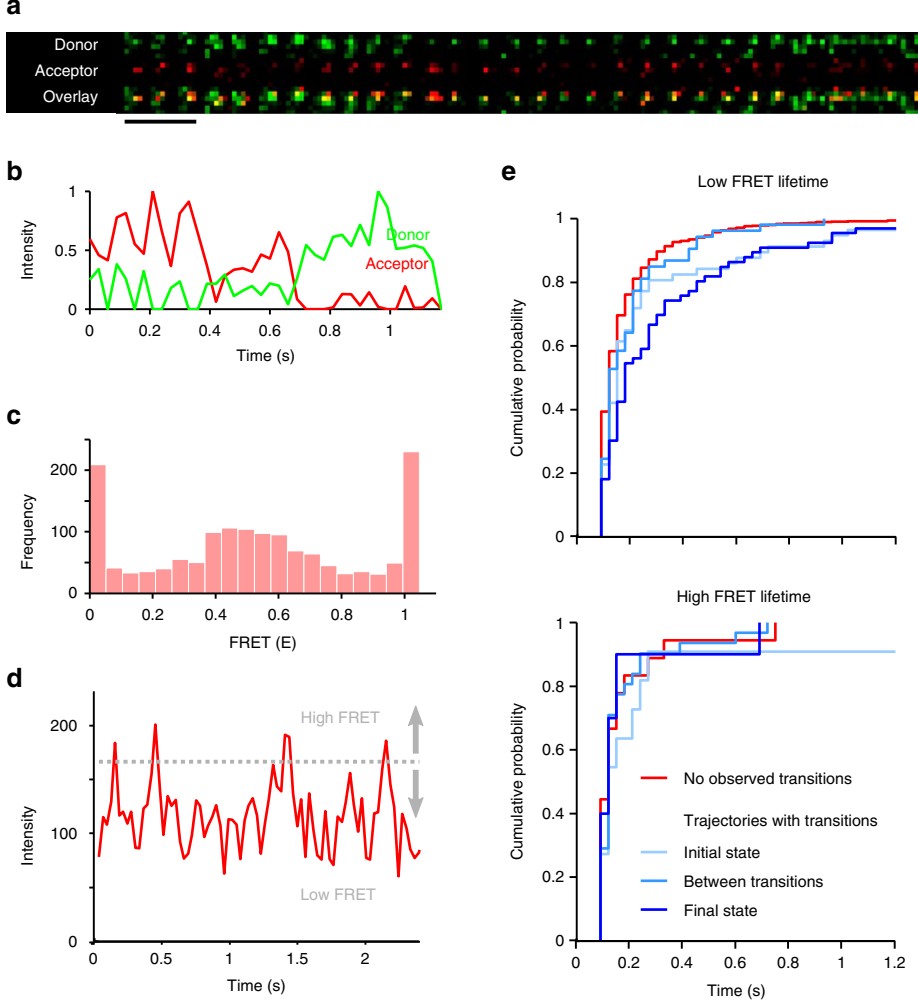

**Fig. 4** Single-molecule FRET dynamics. **a** Sequential frames (30 ms per frame) of part of a single NTS1 dimer trajectory observed by single-molecule FRET. Donor (green) and acceptor (red) channels together with their overlay (registered to nearest pixel) show temporal fluctuation in FRET in the dimer species (note, relative contrast is set to illustrate both channels in overlay). Scale bar: 5 μm. See also Supplementary Movie 5. **b** Representative NTS1 dimer fluorescence traces (green: donor, red: acceptor, see also Supplementary Fig. 9) illustrate temporal fluctuation within a single dimer trajectory; donor and acceptor fluorescence intensities are anti-correlated. **c** FRET efficiency histogram from 40 dimer trajectories shows a broad distribution of FRET efficiencies consistent with high and lower acceptor fluorescence intensity populations, suggesting multiple dimerisation interfaces. **d** Fluorescence intensity of the acceptor in a dimer species is shown for a single dimer trajectory (see also Supplementary Fig. 8). A bimodal acceptor fluorescence intensity distribution, showing high- and low-FRET states, was measured for NTS1 dimers in single-molecule experiments (Fig. 2b and Supplementary Fig. 8). Rather than observing dimer trajectories with acceptor fluorescence intensities at either low or high levels, the acceptor fluorescence intensity showed changes within trajectories, suggesting that the bimodal acceptor fluorescence intensity distribution (Fig. 2b) is a result of dynamic interconversion between different dimerisation interfaces. Interconversion between high- and low-FRET states was analysed. A conservative threshold was defined (166.5 counts—grey dashed line in **d**, see also Supplementary Fig. 8), above which intensities have <1% probability of belonging to the low-FRET state. **e** Lifetime measurements above and below this threshold show dimers exhibiting higher propensity for greater dwell-times in low-FRET (top panel) than in high-FRET (bottom panel) configurations, in keeping with the relative mixing proportions of the two component intensity distributions and FRET efficiency distribution. The lifetime of the low-FRET state is dependent upon it being an initial, mid-, or final transitional state, which could be suggestive of two or more low-FRET states exhibiting differing stability, with the capability to interconvert via a high-FRET state

intermediate protein surface density regime remains inaccessible to either method. Complementary Monte Carlo (MC) simulations were employed to bridge this concentration gap. An estimated mean monomer–monomer collision frequency of ~0.31 collisions $\mu m^{-2} \, s^{-1}$ was calculated from the experimentally determined diffusion rates, and protein surface densities, assuming a particle radius of 2.5 nm (see Supplementary Note 3 and Supplementary Movie 4)[30]. With the experimentally observed dimer formation rate of 0.081 $\mu m^{-2} \, s^{-1}$ (Fig. 3a), this means that ~20% of all monomer–monomer collisions result in dimerisation. Using the experimental diffusion coefficients, the diffusion of monomers at specific surface densities was simulated, ascribing a 20%

probability, based on the estimate of dimerisation collision efficiency, that any random collision (collision diameter <5 nm) would result in dimer formation, upon which the monomers were replaced with a single dimer particle in the simulation. Following dimer formation, our experimental $k_{off}$ (0.58 $s^{-1}$, 95% CI [0.45,1.07] $s^{-1}$) was used to determine the probability of a dimer dissociating back into two monomers. Each simulation was run until dynamic equilibrium and the proportion of dimers relative to the total receptor concentration was determined. The measured dimer population observed in the smFRET experiments could be reproduced in the MC simulations (Fig. 3c, d and Supplementary Fig. 4). At protein densities of ≥10³ $\mu m^{-2}$,

**Table 1 Ensemble interprotomer FRET**

| TM/H (residue) | $E_{cor}$(apo) | n | $E_{cor}$(+NT) | n | ($\Delta E_{cor}$) | Pr($\Delta E_{cor}$ > 0) |
|---|---|---|---|---|---|---|
| 1 (A90C$^{1.58}$) | 0.73 ± 0.03 | 13 | 0.77 ± 0.04 | 9 | −0.038 | 0.73 |
| 2 (Y104C$^{2.41}$) | 0.9 ± 0.1 | 6 | 0.9 ± 0.1 | 6 | 0.054 | 0.40 |
| 3 (C172$^{3.55}$) | 1.03 ± 0.05 | 5 | 1.02 ± 0.02 | 5 | 0.0056 | 0.49 |
| 4 (T186C$^{4.42}$) | 0.77 ± 0.03 | 13 | 0.78 ± 0.04 | 13 | −0.010 | 0.57 |
| 5 (A261C$^{5.52}$) | 0.98 ± 0.06 | 6 | 1.05 ± 0.09 | 6 | −0.073 | 0.70 |
| 6 (V307C$^{6.34}$) | 0.99 ± 0.04 | 11 | 0.82 ± 0.04 | 11 | 0.17 | 0.99 |
| 7 (L371C$^{7.55}$) | 0.68 ± 0.07 | 6 | 0.7 ± 0.1 | 6 | −0.073 | 0.67 |
| 8 (Q378C$^{8.52}$) | 0.89 ± 0.02 | 8 | 0.89 ± 0.02 | 8 | 0.0047 | 0.43 |

The mean corrected FRET efficiency ($E_{cor}$) for n replicate experiments, and standard error of the mean are given for NTS1 labelled on TM1–7 or H8 reconstituted in brain polar lipid liposomes with (+NT) or without (apo) 5 μM agonist neurotensin. The difference between the mean corrected FRET efficiency for the receptor in the presence and absence of agonist ($\Delta E_{cor}$), and the probability (Pr) that $\Delta E_{cor}$ is larger than zero is given

simulated dimeric populations of ~86%–90% are predicted, in agreement with those observed in earlier ensemble FRET experiments on NTS1 (~89%)[21]. It is also notable that dimerisation events become increasingly rare during simulations below receptor densities that might be expected in vivo (Fig. 3c), in agreement with our experimental results (Supplementary Fig. 4a). This observation may provide an explanation for conflicting reports on GPCR dimerisation in different systems.

**Dimerisation interface**. The receptor interface at which the monomer–monomer collisions occur was not specifically taken into account in the MC simulations, although its effect is likely to be accounted for in the collision efficiency of 20% estimated from the experimental observations. A simple model in which the receptor has four distinct faces with only one combination of these compatible with dimerisation would yield an expected maximum dimerisation probability of ~6% in the event that all of these collisions would be productive, which is unlikely. The higher dimerisation efficiency observed suggests that the interface of dimerisation is likely to be less stringently defined than a single productive interface.

Indeed, a multimodal fluorescence intensity distribution representing high- and low-FRET states was observed for the acceptor intensity in NTS1 FRET dimers in the single-molecule experiments (Fig. 2b), consistent with multiple dimeric states. To quantify this, a Gaussian mixture model containing one to four components was fitted to the logarithm of the intensity distribution and the relatively likelihood of each model was assessed from the Bayesian information criterion (BIC) value of each model, where higher BIC values indicate more probable models (Supplementary Fig. 6). The intensity distribution of the acceptor FRET spots was not well described by a single component, in contrast to the intensity distribution of the donor (Fig. 2a and Supplementary Fig. 6i); a two- or three-component model, representing a combination of high and lower energy transfer states, described the intensity distribution of the acceptor FRET spots better than the single component model (Supplementary Figs. 6g and 7). The acceptor intensity distribution of a control sample in which NTS1 monomers were doubly labelled with Cy3 and Cy5 to measure intramonomer FRET (Fig. 2c) was well described by a single component (Supplementary Fig. 6h), suggesting that the observation of multiple components in the acceptor intensity distribution of the dimeric species (Fig. 2a) is the result of at least two populations with different dimer configuration with distinguishable acceptor intensities (i.e., FRET efficiencies).

To determine the nature of these populations, the dimer acceptor trajectory intensities were analysed (Fig. 4 and Supplementary Note 6) in order to categorise dimers as belonging to either high- or low-FRET dimer configurations. The acceptor intensity distribution was equally compatible with a two- and

three-component model (Supplementary Fig. 6g), with one high and either one or two lower energy transfer populations, respectively (Supplementary Fig. 7). Here, the simplest two-state model was chosen in order to aid assignment of the acceptor intensity to a specific subpopulation. A conservative intensity threshold was defined to describe the high acceptor intensity state (Fig. 4d, Supplementary Fig. 8, and Supplementary Note 6), above which intensities had <1% probability of belonging to the low-FRET state population. Rather than observing the presence of two distinct trajectory intensity populations, the minor high-FRET component was found to originate from short-lived fluctuations in acceptor intensity (i.e., threshold crossing, Fig. 4d, and Supplementary Fig. 8). Anti-correlated donor and acceptor intensities from dimer trajectories (Fig. 4b, and Supplementary Fig. 9) illustrated rapid temporal fluctuations that could also be observed in overlaid two-colour image sequences (Fig. 4a and Supplementary Movie 5). Within the overall FRET efficiency distribution (Fig. 4c) the contribution of high FRET efficiency observations (E > 0.9) was 20.4%, in close agreement with the 16% fractional contribution attributed to high acceptor intensity states in the acceptor intensity distribution (Fig. 2b).

Dimer acceptor trajectories were analysed for threshold crossing and dwell-time in high- and low-intensity states was extracted (Fig. 4e). Dimers exhibited greater dwell-time in low-intensity conformation states, in keeping with the relative mixing proportions of the two-component distribution, indicative of a more stable lower intensity state with brief forays into a higher-FRET conformation. Cumulative lifetime probability of low-FRET states reveals an apparent divergence of dwell-time in the low-FRET state dependent upon whether this low-state conformation is adopted as the initial, mid-, or final recorded transition of the dimer trajectory (Fig. 4e, top panel). This observation could be explained by the presence of two or more low-FRET states of differing stability and propensity of formation capable of interconversion via a high-FRET state intermediate (Supplementary Note 6).

To further probe the dimerisation interface, NTS1 was singly labelled with spin labels (for DEER) or fluorescent probes (for ensemble FRET) at the intracellular end of each TM and H8 (specifically residues 90$^{1.58}$, 104$^{2.41}$, 172$^{3.55}$, 186$^{4.42}$, 261$^{5.52}$, 307$^{6.34}$, 371$^{7.55}$, and 378$^{8.52}$, see Supplementary Fig. 1) and reconstituted in brain polar lipid (BPL) liposomes. Intradimer distances were probed between corresponding TMs in each protomer (i.e., TM1–TM1, TM2–TM2 etc.), to assess the relative proximity of each TM to the dimerisation interface.

DEER revealed broad distance probability distributions spanning ~4 nm with multiple peaks for all intradimer TM–TM distances (Fig. 5 and Supplementary Fig. 10), unlikely to be solely due to the conformational spread of the spin label tethers,

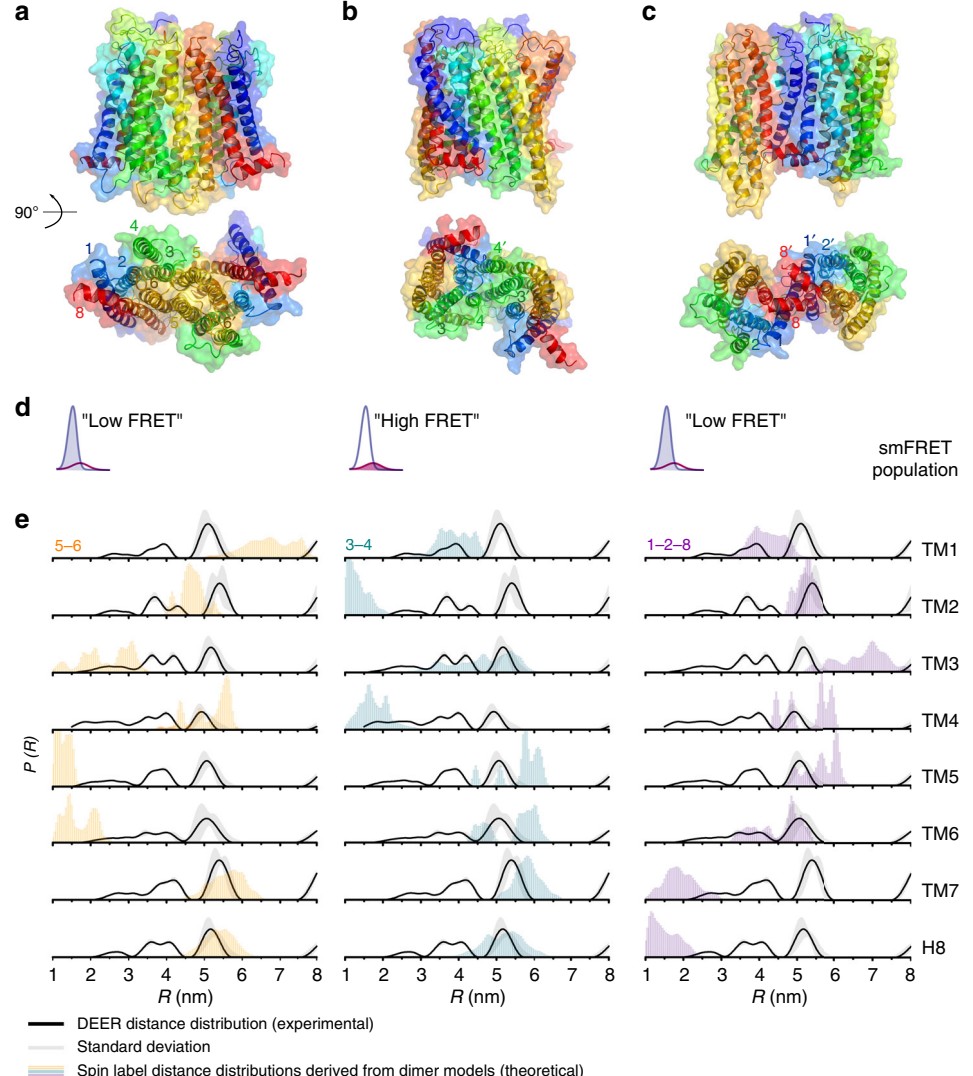

**Fig. 5** Proposed oligomerisation interfaces. Linear combinations of a set of seven structural models of NTS1 dimers combining NTS1 crystal structure monomers (PDB 4BUO[26]) with GPCR dimer crystal structure templates (Supplementary Fig. 12) were compared to the experimental FRET results (for apo receptor). A combination of mostly **a** a TM5-6 interface (long TM4-4 distance) with a small contribution of a **b** TM3-4 (short TM4-4) and/or a **c** TM1-2-H8 interface (long TM4-4), was most consistent with the ensemble FRET results (see also Supplementary Table 3). **d** For each model, the smFRET population (see Fig. 2b) most consistent with the estimated average TM4 inter-residue distance for that model is indicated. **e** Predicted DEER distance distributions calculated for each of the models taking into account the spin label rotamers (see Supplementary Fig. 1c) are plotted here as histograms (5–6 in orange, 3–4 in green, and 1–2–8 in purple) on top of the experimental distance distributions for apo receptor (black line, with grey shaded area indicating the fitting error, see also Supplementary Fig. 10)

suggestive of multiple dimer configurations, in agreement with the smFRET data (Fig. 4 and Supplementary Fig. 6).

Interprotomer resonance energy transfer efficiencies ($E_{cor}$) from complementary ensemble FRET experiments are given in Table 1. The average TM4–TM4 $E_{cor}$ of $0.77 \pm 0.03$ (Table 1) corresponds to an inter-label distance of $5.7 \pm 0.3$ nm (assuming $\kappa^2 = 2/3$, and using $R_0 = 7.0$ nm for the Förster radius of the A488–A555 donor and acceptor pair[31]), which agrees well with the TM4-4 inter-monomer distance estimated from smFRET (~5 nm, see Supplementary Note 5 and Supplementary Fig. 11). Additionally, statistically significantly lower transfer efficiencies (i.e., longer distances) were found for NTS1 labelled on TM1, 4, and 7 compared to TM2, 3, 5, 6, and H8 (Table 1 and Supplementary Table 2). Although the IC ends of TM1 and 7 are indeed in close proximity in the NTS1 protomer (Supplementary Fig. 1) and similar FRET efficiencies can therefore be expected for labels on these positions, the IC ends of, e.g., TM2 or H8 and TM5 or 6 are

on the opposite sides of the molecule, and the observation of high FRET efficiency for labels at all of these sites does not appear consistent with a single dimerisation interface. While this discrepancy could in part be explained by the flexible linkers attaching the fluorophores (~1.5 nm), it could also be the result of an ensemble of dimer interfaces where different groups of TM helices can be present at the interface in different dimer configurations, consistent with the smFRET and DEER results.

Taken together, these findings provide strong evidence that NTS1 dimers explore a number of metastable interfaces. To identify interfaces compatible with the experimental observations, structural NTS1 dimer models were constructed representing four different classes of interfaces observed in GPCR crystal structures comprising: (1) TM1–2–H8 (ß1AR, PDB 4GPO; μOR, PDB 4DKL; and κOR, PDB 4DJH); (2) TM3–4 (H1R, PDB 3RZE); (3) TM3–4–5 (CXCR4, PDB 3OE0; and ß1AR, PDB 4GPO); and (4) TM5–6 (μOR, PDB 4DKL, Supplementary Fig. 12)[32–36]. All seven

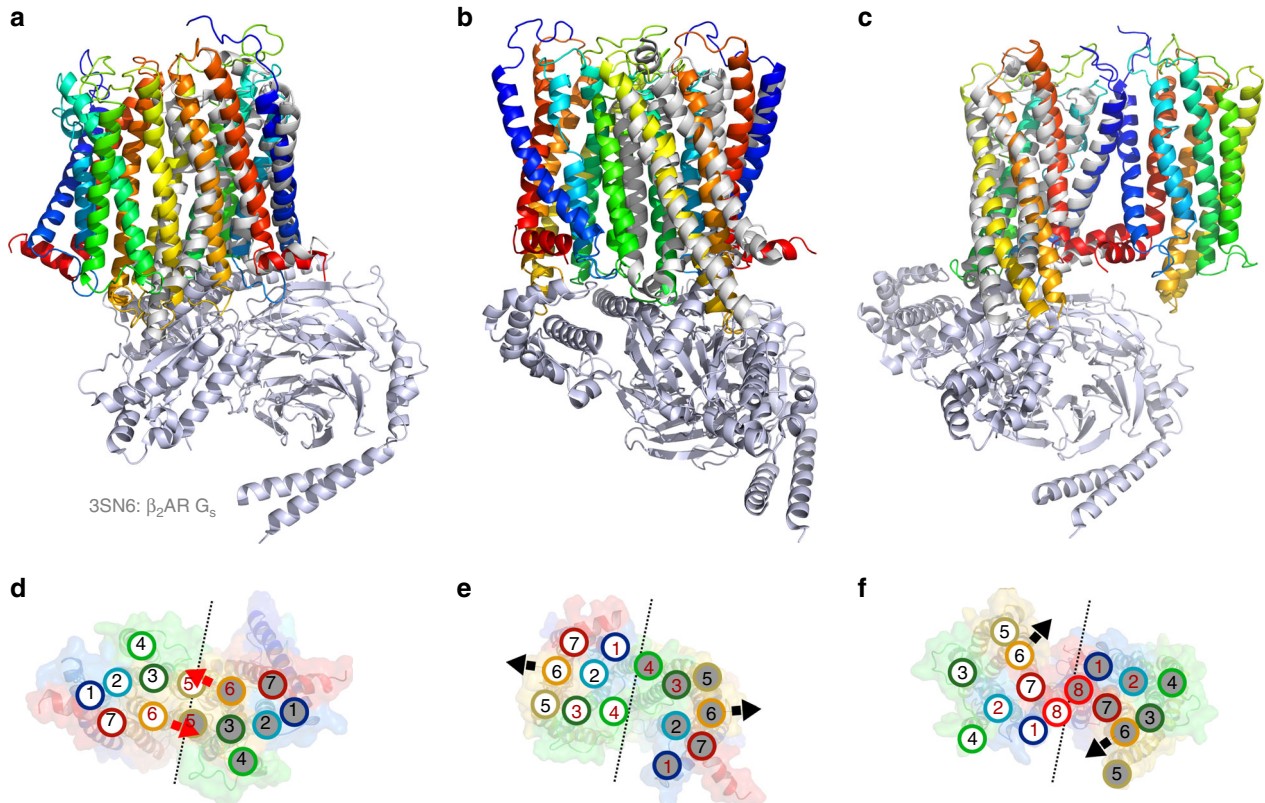

**Fig. 6** Alignment of G protein-bound receptor structure to MD dimer models The crystal structure of the active-state β₂-adrenergic receptor in complex with the heterotrimeric G_s protein (grey; PDB 3SN6, T4 lysozyme and nanobody are omitted here)[41] aligned to the **a** TM5-6, **b** TM3-4, and **c** TM1–2–H8 dimer interface models of NTS1 is shown here with schematic representations of the intracellular view of each model in panels **d**–**f**, respectively. Outward movement of TM5 and 6 in the active receptor would likely result in clashes between the two protomers of the TM5–6 interface dimer (**a**, **d**), but not for the other dimer models

dimerisation interfaces were stable during 10 μs coarse grained (CG) simulations in BPL-like bilayers (see Methods section), and are consistent with earlier proposed GPCR oligomerisation models based on competition studies with synthetic peptides[37], cross-linking experiments[38], and CG simulations[39] (Supplementary Table 1).

The CG simulation frames for the seven dimer interface models were converted back to atomistic representation[40]. The inter-residue distances along the trajectories were measured for all labelling sites used in the ensemble FRET and DEER measurements, and the corresponding average estimated FRET efficiencies were calculated for each labelling site for each model. Given the evidence supporting multiple NTS1 dimer interfaces, the resulting estimated average FRET efficiencies for all possible linear combinations of these interface models (including between one and all seven models) were compared to the experimental FRET results, using least-squares optimisation (see Methods section). The relative likelihood of the different combinatory models was assessed from the fit residuals (RSS) and the Akaike information criterion (AIC) value of each model (Supplementary Table 3).

A combination of a TM5–6 interface with a small fractional contribution ($f$) of either a TM3–4 ($f_{56} = 0.8 \pm 0.3$, and $f_{34} = 0.2 \pm 0.3$, respectively) or a TM1–2–H8 ($f_{56} = 0.9 \pm 0.2$, and $f_{128} = 0.1 \pm 0.2$, respectively) interface gave the best two-component fit to the experimental ensemble FRET results (Fig. 5a–c, Supplementary Table 3). A combination of these three interfaces ($f_{56} = 0.8 \pm 0.3$, $f_{34} = 0.04 \pm 0.4$, and $f_{128} = 0.1 \pm 0.6$) produced a better fit (i.e., lower RSS), but was a less probable model based on its higher AIC value (Supplementary Table 3). A single-component model with either a

TM5–6 interface, or a TM3–4 interface produced a worse fit (i.e., higher RSS), but was equally probable based on AIC as the two-component models (Supplementary Table 3). Thus, in agreement with the evidence from smFRET and DEER, a combination of these dimerisation interfaces would be most probable (Fig. 2a, and Supplementary Fig. 6).

Theoretical DEER distance distributions were calculated for the three dimer models (Fig. 5e, shaded histograms), taking into account the conformational spread of the spin labels (see Supplementary Fig. 1); the individual predicted distance distributions for each of the models show partial overlap with different peaks in the experimental distance distributions (Fig. 5e, solid black line). The calculated distance distributions were compared to the experimental DEER results, using least-squares optimisation, and the RSS and AIC of the different model combinations compared to the experimental DEER data are given in Supplementary Table 4. A linear combination of two or three models gave significantly lower AIC values than any of the individual models. Thus, a combination of multiple interface models is consistent with the observation of both short and long distances for most of the inter-TM DEER measurements, with the combined predicted distance distribution covering most of the experimentally observed distances (Fig. 5). However, the relative contributions of the different peaks in the DEER distance distribution, and thus the relative contribution of the different models obtained from the fits, cannot reliably be interpreted, since the probability for longer distances (>5 nm) cannot accurately be determined from the relatively short DEER traces (Supplementary Fig. 10) suffering from strong relaxation due to the two-dimensional distribution of the spin labels in the

lipid bilayers. The average inter-protomer $C_\beta$–$C_\beta$ distances for the TM4 residue ($168^{4.42}$) labelled in the smFRET experiments, from simulations for the TM5–6 model ($4.6 \pm 0.1$ nm) and the TM3–4 model ($1.5 \pm 0.2$ nm), agree well with the estimated average distances of the low- and high-FRET populations observed in the smFRET experiments, with similar relative fractional contributions ($f_{low} = 0.84$, $f_{high} = 0.16$). The TM1–2–H8 interface model showed a longer average $C_\beta$–$C_\beta$ distance ($5.3 \pm 0.2$ nm) for residue $168^{4.42}$ in the simulations, and is thus unlikely to represent the minor high-FRET population observed in the smFRET experiments (Fig. 5e), but could contribute to the major low-FRET population.

**Ligand modulation of dimerisation interface**. Addition of saturating amounts of agonist (NT, 5 μM) had only a statistically significant effect on the FRET efficiency observed for TM6, where NT showed a 99% probability of decreasing $E_{cor}$ (Table 1 and Supplementary Table 2), reflecting an increase in the inter-TM6 distance in the dimer. Considering the consensus that TM6 undergoes the largest conformational change upon GPCR activation and G protein coupling[41], it is possible that the observed agonist-induced change in $E_{cor}$ is due to a conformational change in TM6 of the protomers in the dimer as typically observed in the monomer, with the conformational change giving rise to the change in $E_{cor}$ directly, and/or indirectly by promoting a change in the preferred dimerisation interface. Although the physiological significance of NTS1 homodimerisation has not yet been studied in vivo, the observation that agonist can elicit a conformational change of TM6 in the dimer as in the monomer suggest that certain conformations of dimers could be signalling competent. Alignment of the activated G protein-coupled structure of the $\beta_2$-adrenergic receptor (PDB 3SN6[41]) to the NTS1 dimer models (Fig. 6) suggests that the TM5–6 dimer interface, which is most consistent with the ensemble data for the apo receptor, is not compatible with the canonical outward movement of TM5 and 6 upon receptor activation, thus precluding receptor activation, as previously discussed[33]. Conversely, the TM3–4 or the TM1–2–H8 dimer interface models could support G protein activation. As for the apo receptor, the best fit of the MD dimer models to the FRET efficiency in the presence of agonist was obtained with a combination of the TM5–6 and TM3–4 interface models (Fig. 5a, b, and Supplementary Table 3). However, a slight decrease in fractional contribution of the TM5–6 interface was found ($f_{56} = 0.7 \pm 0.3$, and $f_{34} = 0.3 \pm 0.3$ in the presence of agonist compared to $f_{56} = 0.8 \pm 0.3$, and $f_{34} = 0.2 \pm 0.3$ for the apo dimer), and shown to be statistically significant by an F-test, suggesting that receptor activation could lead to a change in the dimerisation interface, possibly promoted by the outward movement of TM6 (Fig. 6). However, it must be noted that the models used represent the inactive state of the receptor and the receptor in an active conformation may yield dimer interfaces not included in the present study.

## Discussion

Homo- and heterodimerisation of GPCRs has increasingly been associated with altered signalling profiles[2]. NTS1 homodimerisation has thus far not been studied in vivo, but has been shown to lower the affinity of G protein nucleotide exchange in vitro, while NTS1 heterodimerisation has been shown to modulate G protein selectivity[42] and has been implicated in lung cancer cell growth[43]. However, the mechanisms through which dimerisation modulates NTS1 function remain unclear.

In agreement with recent observations for other GPCRs[23], the data presented here support the notion of a dynamic transient NTS1 dimer with a half-life ($t_{1/2} = 1.2$ s) consistent with previous reports on other receptors ($\sim$0.1–5 s)[15–17]. Interestingly, NTS1 showed a lower oligomeric population of 9.8% compared to $\sim$30–60% reported previously at similar receptor densities ($\sim$0.5–2 μm$^{-2}$) for other GPCRs[15–17]. This may reflect receptor type-dependent differences in dimerisation propensities, but may also be due to differences in experimental conditions; previous single-molecule studies of GPCR oligomerisation dynamics have relied on co-localisation and stepwise photobleaching, which may also interpret protomer crowding as oligomerisation[15–17]. Here, by using FRET, we are sensitive to monomer separations 10-fold smaller than distinguishable with the previously employed methods, and can thus more reliably determine bona fide protein–protein interactions. Furthermore, here, intrinsic receptor–receptor interactions are being characterised, unperturbed by additional cellular interactions, suggesting that dimerisation is an inherent property of NTS1, which may be driven by hydrophobic mismatch as has been proposed for other GPCRs[44], and the extent of which may be modulated by the cellular environment. The observed $K_d$ for NTS1 dimer formation of 7.1 μm$^{-2}$ (95% CI [6.7,12.3] μm$^{-2}$, Supplementary Note 1) lies within the reported physiological range of average cell surface receptor densities (2–40 μm$^{-2}$)[45]. Thus, possible sequestration (whether temporal or constitutive) of NTS1, leading to changes in local concentrations, would dramatically alter dimerisation rates on the sub-second timescale, with potentially functional implications. Indeed, clustering of GPCRs in microdomains has previously been reported to affect GPCR dimerisation and signalling[46].

While NTS1 reconstitution in liposomes has previously been shown to yield symmetric reconstitution with $\sim$50/50 distribution of both orientations[21], the reconstitution orientation in the droplet interface bilayers used in the smFRET experiments is unknown. Symmetric reconstitution would at most halve the effective protein density, since each receptor will only be able to interact with half of the receptors present, resulting in an underestimation of the dimer half-life, $t_{1/2} = 1.2$ s, (95% CI [0.65,1.54] s), and the $K_d = 7.1$ μm$^{-2}$ (95% CI [6.7,12.3] μm$^{-2}$) by a factor of two. Whilst we cannot exclude symmetric reconstitution, or the formation of non-physiological antiparallel dimers in our experimental system, the observed dimer half-life (1.2 s) is consistent with previous reports on other GPCR dimers ($\sim$0.1–5 s)[15–17]. Furthermore, the inter-label distance in the high-FRET state is too short to originate from an antiparallel dimer, and high- and low-FRET states were observed to interconvert (Fig. 4), precluding that antiparallel dimers contribute to the measured smFRET. Thus, the effect of reconstitution orientation on our results is expected to be modest (see also Supplementary Note 7).

To our knowledge, here, we demonstrate the first evidence for a GPCR dimer "rolling interface" in which monomers sample a number of different configurations that can interconvert during the dimer lifetime by rotation of the monomers relative to each other. Many GPCR dimers have been proposed to adopt a variety of (conflicting) configurations using different approaches with most, if not all, TM domains having been implicated in the interaction interface (see Supplementary Table 1). The rolling interface model accommodates these previously contradictory findings as a consequence of the dynamic nature of the dimer interface, whilst demonstrating that a number of preferred arrangements exist. Although multiple dimerisation interfaces could also lead to the formation of higher-order oligomers as observed for other GPCRs[17], higher-order oligomers were previously excluded for NTS1 in this experimental ensemble system (at $\sim$1–3 $\times$ 10$^3$ μm$^{-2}$)[21] and are not detected here at the single-molecule level at much lower densities ($\sim$0.4 μm$^{-2}$), with MC simulations unifying these conditions (Fig. 3c). However, formation of higher-order oligomers at even higher protein densities

($>10^4\,\mu m^{-2}$) well above average physiological expression levels cannot be excluded.

The possibility of multiple dimer interfaces has been previously proposed based mainly on evidence from cross-linking studies[47,48], and is in line with recent computational work[39,49,50]. Changes in the dimerisation interface have previously been shown to be linked to receptor activation[48,50]. Here, we extend the model, showing that apo NTS1 forms a metastable dimer, which samples multiple dimer configurations with distinct dimerisation interfaces with varying stability on a per molecule basis. Given the sequence similarity with other GPCRs in the TM domain, and previous observation of multiple dimerisation interfaces for both class A and C GPCRs, the "rolling interface" model may also hold for other receptors in the superfamily. At a physiological level, the presence of a number of metastable interfaces increases the likelihood of dimer formation through random collisions. Furthermore, the functional consequence of dimerisation may depend on the interface involved in the association, and dynamic reorientation of the dimerisation interface provides an increased level of biological complexity, which could play a role in biased signalling or other forms of allosteric regulation. Indeed, heterodimerisation of the κ-opioid receptor with NTS1 biased its signalling toward the β-arrestin pathway[42]. The dimer configuration may influence the ability of the protomers to undergo conformational changes associated with G protein signalling as suggested by different dimer models (Fig. 6). Our findings, as well as previous cross-linking[7,38] and computational studies[39], suggest that the activation state of the receptor affects the preferred dimerisation interface. Here, apo NTS1 appears to favour a TM5-6 interface that is incompatible with G protein coupling. It is thus tempting to speculate that dimerisation might also have evolved as a mechanism to increase the fidelity of GPCR signalling by lowering basal signalling in the absence of agonist stimulus. Such a mechanism would be consistent with the previously reported very low basal signalling of NTS1[51], and the lower potential of NTS1 dimers compared to monomers to catalyse G protein nucleotide exchange in detergent[23]. Alternatively, intradimer interactions may affect the conformation of the protomers, which in turn may affect G protein signalling properties including specificity[52].

It has to be noted that the relevance of the crystallographic dimers used here as templates for specific dimerisation interfaces is contentious. Indeed, many GPCRs have also been crystallised in antiparallel "dimeric" configurations. Nevertheless, in the absence of other high-resolution structural information on GPCR dimers, the energetically favourable parallel interfaces observed in crystallography may represent viable models of dimerisation interfaces in vivo, although other, including asymmetric interfaces, may exist.

As previously argued, e.g., by Bouvier and Hebert[53], transience of dimerisation as observed here for NTS1 and previously for other GPCRs[13–18] does not preclude any role in modulation of signal transduction, and may in fact be an important feature of its role in signal regulation. GPCRs show complex signalling patterns, activating different G proteins as well as arrestins, requiring significant conformational plasticity. A combination of multiple dimerisation interfaces and concentration-dependence of dimerisation could provide an attractive speculative mechanism for cells to dynamically influence signalling cascades rapidly and in a temporally and spatially regulated manner, e.g., through stabilisation of particular dimer interfaces by the local lipid environment[49] or through modulation of receptor density governing dimer fraction. This could have broader implications for regulation of GPCR signalling, extending the traditional ligand-dependent view of functional selectivity to incorporate direct contributions from allosteric receptor–receptor interactions, analogous to the effect of RAMPs on Class B GPCRs[54].

Modulation of dimer lifetime, preferred interface, or interface interconversion with possible implications for signalling could occur endogenously, e.g., through the lipid environment[39,49], or by exogenous compounds. Indeed, it has been demonstrated that a variety of GPCRs, including NTS1, are aberrantly overexpressed in cancerous cells[19,55] and, as such, can be envisaged to possess altered dimerisation and signalling profiles in disease states, making the dimerisation interface a potential drug target.

In conclusion, this study provides compelling evidence that NTS1 dimers are a dynamic species which can explore multiple interfaces and contributes to a growing body of work suggesting that class A GPCR dimerisation can be transient and concentration-dependent, providing an explanation for previously conflicting reports on GPCR dimerisation interfaces or dimerisation propensity in different experimental systems. While the exact physiological role of dimerisation is still unknown, a better understanding of its physical and temporal nature will inform future functional studies.

## Methods

**Production and liposome reconstitution of labelled NTS1**. Single mutations were introduced into a Cys-depleted background mutant of rat NTS1 (C172S[3.55], C278S[IC3], C332S[6.59], C386S[C-term], and C388S[C-term]) using the QuikChange II protocol (Stratagene): A90C[1.58] (TM1), Y104C[2.41] (TM2), S(C)172C[3.55] (TM3, re-introduced native Cys), T186C[4.42] (TM4), A261C[5.52] (TM5), V307C[6.34] (TM6), L371C[7.55] (TM7), and Q378C[8.52] (H8). Primer sequences are shown in Supplementary Table 5. Labelling sites were chosen based upon homologous sites previously employed in a site-directed spin labelling study of rhodopsin[56]. Microscale thermophoresis measurements were performed to test ligand and G protein binding to the (labelled) NTS1 cysteine mutants (see Supplementary Note 8). NTS1 was expressed in E. coli BL21(DE3) (competent cells purchased from Agilent Technologies), as a fusion construct NTS1BH₆ (MBP-TEV-rT43NTS1-His₆-TEV-TrxA-His₁₀), where NTS1 is truncated at the N-terminus (1–42), has a hexa-His-tag added to its C-terminus, and is flanked by TEV protease recognition sites separating it from its N- and C-terminal fusion partners, maltose binding protein and thioredoxin, respectively, followed by an additional C-terminal deca-His-tag. Starter cultures (5–7.5 mL LB, 1% (w/v) glucose, 100 μg/mL ampicillin) were inoculated with a single colony of NTS1BH₆ plasmid-transformed E. coli BL21 (DE3) each, and incubated overnight at 37 °C and 200 rpm. In 2 L conical flasks, aliquots of 500 mL 2xYT medium supplemented with 0.2% glucose, and 100 μg mL⁻¹ ampicillin were inoculated with 5 mL of starter culture each, and incubated at 37 °C at 200 rpm, usually growing 10–20 L of culture at a time. When the cultures reached an OD₆₀₀ of 0.3 the temperature was decreased to 26 °C, until the cultures reached an OD₆₀₀ of 0.5, at which point expression was induced by addition of IPTG to a final concentration of 0.25 mM. Cells were harvested by centrifugation (7000 g, 4 °C, 15 min) after overnight expression and either solubilised directly or flash-frozen in liquid nitrogen and stored at −80 °C. NTS1BH₆ was purified by immobilised metal affinity and ligand affinity chromatography. All purification steps were carried out at 4 °C unless stated otherwise. Specifically, cell pellet was resuspended in 2 mL per gram of cell pellet 2 × solubilisation buffer (100 mM Tris pH 7.4, 400 mM NaCl, 60% glycerol) supplemented with protease inhibitors (2 μg mL⁻¹ leupeptin, 2 μg mL⁻¹ pepstatin A, and 3 μg mL⁻¹ aprotinin). Cells were incubated for 20 min with 1 mg of DNase I and 1 mg mL⁻¹ lysozyme, and subsequently lysed using a French press, passing cells 2–3 times through the press at 16 kpsi. DDM, CHAPS (both Melford) and CHS (Sigma-Aldrich) were added dropwise to the lysate under stirring to a final concentration of 1%, 0.5%, and 0.1% (w/v), respectively. MilliQ H₂O was added to give a final volume of 4 mL per gram of pellet, and the cells were left to stir for 6 h. Unsolubilised material was pelleted by centrifugation (70,000 g, 4 °C, 60 min). The solubilised fraction (supernatant) was filtered through a 0.2 μm syringe filter and imidazole (Merck) was added to a final concentration of 50 mM, before loading the sample onto a freshly charged, 5 mL HisTrap HP column (GE Healthcare), pre-equilibrated with NiA buffer (50 mM Tris-HCl pH 7.4, 200 mM NaCl, 10% glycerol (v/v), 0.5% CHAPS (w/v), 0.1% DDM (w/v), 0.1% CHS (w/v), 50 mM imidazole, protease inhibitors). The column was washed with 35–40 CV of NiA buffer and the protein was eluted with NiA buffer supplemented with 500 mM imidazole. A₂₈₀ was monitored, the peak fractions were pooled, and the fusion partners were removed by incubation with TEV protease (produced in-house) at a 1:1 molar ratio and 5 mM DTT for ~16 h, at 4 °C. NTS1 was then diluted five-fold with NT0 buffer (50 mM Tris pH 7.4, 10% glycerol, 0.1% DDM (w/v), 0.01% CHS (w/v)) to reduce the NaCl concentration and further purified by ligand affinity chromatography on an NT column, using N-terminally Cys-derived NT (Alta Bioscience) immobilised on Ultralink iodoacetyl resin (Pierce, Thermo Scientific), ensuring the final sample contained only properly folded receptor, capable of ligand binding. Specifically, the diluted nickel column eluate was incubated with approximately 1–2 mL of resin for 2–3 h at 4 °C on a rotating wheel. NT resin was washed with 50–70 CV of NT70 buffer (70 mM

NaCl), followed by a 50 CV wash with NT150 buffer (150 mM NaCl), after which the sample was eluted with NT1 buffer (1 M NaCl) and supplemented with 5 mM DTT. NT column eluate was concentrated by combining several batches on a 1 mL HisTrap HP nickel column (GE Healthcare). The sample was loaded onto the column at 0.5–1 mL min$^{-1}$. Flow-through was collected and re-applied to the column. The column was washed with 100 CV NiA0 buffer (50 mM Tris-HCl pH 7.4, 200 mM NaCl, 10% glycerol (v/v), 0.1% DDM (w/v), 0.01% CHS (w/v)) and NTS1 was eluted in a small volume with 400 mM imidazole.

The eluate was labelled with either (1) Alexa Fluor 488 (A488)/Alexa Fluor 555 (A555, Life Technologies), (2) Cy3/Cy5 (GE Healthcare), or (3) (1-oxyl-2,2,5,5-tetramethylpyrroline-3-methyl)-methanethiosulfonate (MTSL, Toronto Scientific Chemicals), for ensemble FRET, smFRET, or DEER experiments, respectively. Labelling was performed by incubation of NTS1 with a 2–20 times molar excess of label for 5–60 min at room temperature. Excess label was removed by gel filtration (2–3.5 mL HiTrap Desalt columns, GE Healthcare, connected in series, or Zeba Spin Desalting Columns, MWCO 40 K, Pierce, Thermo Scientific), simultaneously buffer exchanging the receptor into 50 mM Tris-HCl pH 7.4, 50 mM NaCl, 1 mM EDTA, 0.1% (w/v) DDM, 0.01% (w/v) CHS, and 10% (v/v) glycerol.

The average labelling efficiency of the fluorophore-labelled samples was determined by comparing the absorption at 280 nm with that of the fluorophore maximum, correcting for the contribution of the fluorophore at 280 nm

$$E_{labelling} = \frac{[Label]}{[NTS1]} = \frac{A_{label}/\varepsilon_{label}}{(A_{280} - CF \times A_{Label})/\varepsilon_{NTS1}} \quad (1)$$

where $A_{Label}$ and $A_{280}$ are the maximum absorption of the fluorophore and the absorption at 280 nm, respectively, $\varepsilon_{Label}$ the extinction coefficient of the fluorophore at its maximum, and $\varepsilon_{NTS1}$ that of NTS1 at 280 nm (56,840 M$^{-1}$ cm$^{-1}$). CF is a correction factor to account for the contribution of the fluorophore at 280 nm which is determined from the excitation spectrum of the free fluorophore in solution as the percentage absorption at 280 nm compared to its maximum absorption. For DEER and ensemble FRET measurements, NTS1 was reconstituted in brain polar lipid (BPL, Avanti polar lipids) liposomes, using Bio-Beads (Bio-Rad) for detergent removal[21]. Specifically, BPL in chloroform was dried down to a lipid film using a rotary evaporator and further dried overnight in a desiccator under vacuum and stored at 20 °C. The lipid film was suspended in liposome buffer (50 mM Tris-HCl pH 7.4, 50 mM NaCl, 1 mM EDTA, saturated with N$_2$) to give a final concentration of 5 mg mL$^{-1}$, and sonicated (3 × 1 min) using a bath sonicator, followed by ten freeze-thaw cycles using liquid nitrogen and a 37 °C water bath. The liposomes were then extruded through a 100 nm polycarbonate filter using a mini-extruder, for at least 11 passes to obtain a homogeneous distribution of liposomes of 100 nm in diameter. DDM was added to the lipid suspension at a final concentration of 0.25% (w/v) and the lipids were gently stirred for 1–3 h. The detergent-liposome mix was then added to the receptor at the desired lipid-to-protein ratio, and incubated for 1 h at 4 °C. Bio-Beads were washed with methanol, followed by MilliQ water, and equilibrated with liposome buffer, and then added at 0.3 g mL$^{-1}$ (wet weight) and the sample was left to incubate overnight on a rotating wheel at 4 °C. Bio-Beads were then removed, and proteoliposomes were harvested by centrifugation (~100,000 g, 3 h, 4 °C). For FRET experiments donor and acceptor-labelled NTS1 was reconstituted together at a 1:1 molar ratio, as well as separately for donor-only, and acceptor-only controls. Initial lipid-to-protein ratios of 1:6000, 1:12,000, and 1:1500 (mol:mol) were used for FRET, donor/acceptor-only FRET control, and DEER samples, respectively. Final lipid-to-protein ratio was determined by sucrose density gradient centrifugation (see Supplementary Note 10 and Supplementary Table 6).

**Droplet interface bilayer single-molecule FRET experiments.** Droplet interface bilayers (DIBs) are formed following the self-assembly of lipid monolayers at water–oil interfaces, with subsequent contacting of two such interfaces giving rise to a bilayer[27]. Here, a solution of lipid in oil is placed on top of a hydrogel and allowed to equilibrate. An aqueous droplet is then introduced in this solution and allowed to come into contact with the hydrogel surface. A lipid bilayer spontaneously forms at the interface between the two aqueous phases. Protein inserts into the bilayer following the incorporation of detergent purified NTS1 in the aqueous droplet. The planar nature of DIBs formed on a thin (<100 nm) hydrogel layer make them amenable to single-molecule imaging via TIRF microscopy (Supplementary Fig. 1). Hydration of the underlying agarose layer enables free diffusion of membrane components[57]. DIBs were prepared following previously reported methodologies[27]. Briefly, molten agarose in water (0.75% w/v, 90 °C) was spin-coated (3000 rpm, 30 s, using a Laurell Technologies Corporation spin coater) on top of a O$_2$ plasma-cleaned glass coverslip prior to incorporation of the coverslip into a poly(methyl methacrylate) (PMMA) micro-device. The agarose-coated coverslip forms the optically accessible base layer upon which the DIB is formed. The micro-machined PMMA device comprises a planar fluidic channel in contact with the underlying agarose-coated coverslip; this channel interdigitates an array of vertical open wells allowing access to the spin-coated agarose layer from the top of the device. A molten solution of agarose (1.8% w/v, 90 °C) in 50 mM NaCl, 1 mM EDTA, 50 mM Tris-HCl at pH 7.4 is pipetted into the fluidic channel such that the agarose fills the full volume of the channel, contacting the underlying spin-coated agarose, providing a source of hydration to the low-volume spin-coated agarose layer. The rehydrating agarose does not enter the wells of the device, maintaining

an accessible area of suitably thin, yet hydrated, agarose for evanescent field penetration in subsequent TIRF imaging. The agarose layer within each well was submerged in oil solution (hexadecane:silicone oil AR20 (19:1)) containing dissolved lipid (1,2-diphytanoyl-sn-glycero-3-phosphocholine (DPhPC) 8 mg mL$^{-1}$) and allowed to equilibrate for 15 min. A ~0.02 μL aqueous droplet (50 mM NaCl, 1 mM EDTA, 50 mM Tris-HCl at pH 7.4) containing Cy3 and Cy5 labelled NTS1 T186C$^{4.42}$ (provided from protein preparations at 0.15 μM (Cy3) and 0.18 μM (Cy5) in 50 mM NaCl, 1 mM EDTA, 50 mM Tris-HCl at pH 7.4 with 0.1% (w/v) DDM, 0.01% (w/v) CHS, 10% (v/v) glycerol, added immediately prior to droplet preparation), was incubated in a separate chamber containing the same lipid in oil solution for 20–25 minutes before being transferred by pipette to the experimental device. Bilayer formation proceeded upon the droplet sinking to the bottom of the oil-containing well and contacting the underlying agarose surface. NTS1 was incorporated in the droplet at a total concentration of 550 pM with a labelling ratio of 100:405:45 Cy3:Cy5:unlabelled. This concentration ratio ensured a greater prevalence of FRET capable dimerisation compared to equimolar labelling conditions, affording the measurement of an increased number of dimers, whilst retaining single-molecule resolution in both donor and acceptor channels. Unlabelled receptor was present as a consequence of incomplete labelling of the acceptor population (10% unlabelled). This was accounted for in all subsequent calculations (see Supplementary Information).

**Microscopy.** A 532 nm laser (LDC-1500, Suwtech, Saxonburg PA, USA) was expanded to a collimated beam diameter of 2 mm before focussing off-axis at the back aperture of a TIRF oil immersion objective lens (60 × Plan Apo N.A. 1.4, Nikon Instruments, UK) mounted on an inverted microscope (Ti Eclipse, Nikon Instruments, UK), giving rise to total internal reflection illumination at the coverslip surface. Emitted fluorescence was collected through the same objective, transmitted through a dichroic mirror (Nikon532 C67195) prior to passing through a long pass emission filter (532 nm EdgeBasic BLP01-532R-25 Semrock, USA) followed by wavelength image splitting to separate out donor and acceptor emission for side-by-side imaging on a single 128 × 128 pixel frame-transfer emCCD detector (iXon DU-860, Andor Technology PLC, Belfast, UK). Wavelength image splitting was achieved with an Optosplit II (Cairn Research, UK) with a dichroic beamsplitter (640 nm FF640-FDi01 Semrock, USA) followed by band-pass filters in each channel (582/75 (FF01-582/75-25 Semrock, USA) and 670/40 (670DF40 Omega Optical, USA)), selected to minimise donor and acceptor emission bleed, with focal parity achieved via a correction lens in the red channel. Images were acquired with a frame time of 30 ms and EM gain of 255 and saved as 16-bit Tagged Image File Format (TIFF) for subsequent analysis. A beam shutter TTL synchronised to the emCCD ensured fluorophore labelled protein experienced no laser exposure prior to image acquisition to avoid photobleaching. Data were acquired from five droplet interface bilayers. Multiple time series acquisitions of 6 s were recorded for each bilayer, with each acquisition in a previously unexposed area of the bilayer. Due to the variability in reconstitution efficiency between bilayers[58] (Supplementary Fig. 4) and the concentration dependency of dimerisation, only bilayers with a Cy3 (donor) receptor density above 0.04 μm$^{-2}$ were considered for further analysis. In the case of NST1 dimerisation, a total of 19 videos were analysed containing 17,853 donor trajectories, comprising 209,712 spots, and 1167 acceptor trajectories, comprising 8561 spots. For the doubly labelled NTS1 construct (A90C$^{1.58}$-T186C$^{4.42}$ labelled with Cy3 and Cy5), smFRET experiments were conducted at a reduced receptor concentration to ensure a predominantly monomeric receptor population; 371 acceptor trajectories, comprising 2816 spots were measured.

**smFRET image analysis.** All data were analysed using ImageJ and Matlab (R2012a) (MathWorks USA). Image stacks were prepared for spot detection and tracking in ImageJ. Background subtraction was performed via the pixel-wise subtraction of time averaged median pixel intensity from each image sequence. Image stacks were subsequently split into donor and acceptor channel images (64 × 128 pixels) with spot detection and particle tracking conducted on each stack using the ImageJ plugin trackmate[59], included in the FIJI distribution of ImageJ. For spot detection an estimated spot size diameter of 3 pixels was employed, with detection following a difference of Gaussians algorithm (DoG) calculating the difference image between two Gaussian smoothed images. This process serves as a bandpass filter removing high spatial frequency noise whilst preserving spatial features on the order of the estimated diffraction-limited spot of a single fluorophore to aid spot detection. Trajectory linking was performed with a maximum linking distance of 5 pixels, a maximum gap-closing distance of 5 pixels at a maximum gap-closing frame gap of two frames. A minimum track length cut-off of three frames was subsequently employed to ensure the acceptance of only bona fide receptor trajectories, excluding any false-positive spot detections. Trajectories were subsequently analysed in Matlab. An image of the TIRF illumination spot following smoothing with a Gaussian blur ($\sigma = 10$ pixels) was used to create an x/y lookup of relative laser illumination intensity to normalise trajectory spot intensity in accordance with the spot position in the TIR illumination area. Mean square displacement and diffusion coefficient of donor and acceptor trajectories was calculated in Matlab following previously reported routines[60]. For the extraction of smFRET efficiencies (E) and the generation of two colour single-spot excised dimer trajectories, donor and acceptor video pairs were individually registered and

overlaid as separate colour channels. Spot detection and tracking was performed on combined images. 40 trajectories lasting longer than 0.5 s and free from spot-crowding in the donor channel (in order to increase reliability of donor spot intensity extraction) were taken forward for the frame-by-frame extraction of smFRET efficiency traces. $x/y/t$ spot trajectory data were used to excise regions of interest from the respective donor and acceptor stacks accounting for image registration offset. Spot fitting and intensity extraction proceeded with 2D Gaussian fitting on the excised donor and acceptor image sub-stacks, respectively. Excised regions of donor and acceptor channels were recombined to create two colour single spot image stacks of FRET dimer trajectories, as illustrated in Fig. 4a. Although sub-pixel resolution was employed for calculating image registration offset and defining spot location, recombined images were simply registered to the nearest pixel, to avoid interpolation and preserve imaging data as acquired.

**smFRET intensity distribution analysis**. Gaussian mixture models were fitted to the logarithm of the single-molecule intensity data for (a) the FRET acceptor intensity of dimeric species; (b) the FRET acceptor intensity of doubly labelled, FRET capable, monomeric protein (control 1); and (c) the direct donor excitation of monomeric singly labelled receptor (control 2), using the Mclust package for R[61]. To account for differences in sample size between the intensity distributions for the three data sets, the data were bootstrapped 1000 times, taking random samples of $n = 1000$ for each of the three data sets. Four Gaussian mixture models composed of 1–4 components were fitted to each bootstrapped data sample, and the corresponding Bayesian information criterion (BIC) value was calculated to assess the relative likelihood of each of the models. To compare between bootstrapped samples, the BIC values for each sample were converted to BIC difference values by subtracting the largest BIC in the set ($BIC_{i,j} - BIC_{i,max}$).

**Monte Carlo simulations**. Two-dimensional Monte Carlo simulations simulating receptor diffusion, dimerisation, and dissociation were coded in Matlab, parameterised by the kinetic parameters determined by single-molecule DIB experiments. Monomeric receptors were described as point-like particles with a collision radius, $r$, diffusing randomly on a 2D surface with diffusion coefficient, $D_{mon}$. These properties were used to determine the time step ($dT$). This was defined as the time period in which under Brownian diffusion a receptor would be expected to on average diffuse a distance equivalent to half a receptor radius. Particle number, surface density and nominal labelling proportion were defined by the user to enable simulation of different experimental conditions. The combined prescribed particle number and density defined the area of the square surface on which the particle diffusion was simulated. 2D periodic boundary conditions were implemented such that the simulated experimental surface approximated a boundary-less continuous surface at the described particle density. Following Einstein's relation, diffusive displacement was assumed to follow a Gaussian probability distribution function with standard deviation given by

$$\sigma = (2D_{mon}dT)^{0.5} \qquad (2)$$

At each time step the random diffusion of each particle was determined in turn by a random-normally distributed step-size, determined by the particle diffusion coefficient with normal distribution described by Eq. 2. The direction of this step was determined by a randomly assigned angle between 0 and 360 degrees. Particle locations were defined with a tolerance of $10^{-5}$. Within a single time step of the simulation each particle was moved sequentially before checking for particle collisions. Collisions were detected by determining the separation between all particles. All particle pairs separated by a distance of equal to, or less than, a receptor collision radius (twice receptor radius) were identified as collisions. Collisions were then determined to proceed to particle dimers with a probability ($p$) described by the estimation based on experimental data using the method described by Hardt[30] (see Supplementary Note 3). Dimer particles diffused with a diffusion coefficient, $D_{dim}$, as measured experimentally, and could dissociate with a probability determined by $k_{off}$ and the duration of the simulation time step. Higher-order oligomers were precluded. Dimer particles maintained their constituent receptors particle identities. Consequently all particles could be categorised by label identity for simulated FRET experiments, where simulated donor and acceptor channel images were generated at 30 ms intervals. Simulations were initiated out of equilibrium with all particles initially described as monomers. Monomer and dimer populations were monitored over time. Dynamic equilibrium was established within 5000 steps in all cases as determined by analysis of a rolling average spanning different time windows for simulations of durations of up to 200 s. For collection of the data in Fig. 3c, 300 particles were simulated at a range of receptor densities spanning the single-molecule, physiological, and ensemble experimental range. A period of 50 s was simulated and simulations run three times at each receptor density. Monomer:dimer population was measured under dynamic equilibrium conditions in each simulation using all time points from the 5001st step onward.

**Ensemble FRET**. NTS1 proteoliposomes (containing 0.15–0.2 nmol receptor) were pelleted (100,000 g, 3 h, 4 °C) and resuspended in 150–200 µL of detergent-free buffer (50 mM Tris-HCl pH 7.4, 50 mM NaCl, 1 mM EDTA, 10% (v/v) glycerol),

yielding a final concentration of ~1 µM. Fluorescence emission spectra were recorded at room temperature on a Perkin-Elmer LS-50B fluorimeter (slit widths 4.5 nm for excitation and 5 nm for emission, to optimise the signal-to-noise ratio). A background sample consisting of empty liposomes was used to correct for any background fluorescence and scattering effects. Spectra were smoothed using Savitzky-Golay filtering (10 nm window, second order polynomial) using OriginPro 8.5 to minimise noise artefacts in the determination of spectral maxima. Apparent FRET efficiencies were determined as described in Supplementary Note 9. The resulting apparent FRET efficiency ($E_{app}$) was further corrected for the donor-to-acceptor ratio ($r_{DA}$), determined from the fluorescence emission spectra, correcting for quenching of the donor emission due to FRET as described by Gordon et al.[62]

$$F_D = F_{ex:D}^{em:D} + F_{FRET,cor} \frac{\Phi_D}{\Phi_A} \qquad (3)$$

where $F_{ex:D}^{em:D}$ is the emission spectrum of the donor in the mixed FRET sample when excited at the donor wavelength, $F_{FRET,cor}$ the FRET signal corrected for bleedthrough and crosstalk as described by Goddard et al.[63], and $\Phi_D$ and $\Phi_A$ the fluorescence quantum yield of the donor and acceptor fluorophores, respectively, which were taken to be 0.92 and 0.10, respectively, as specified by the supplier. As the fluorescence emission is proportional to the sample concentration, multiplied by the quantum yield and the extinction coefficient, the donor-to-acceptor ratio $r_{DA}$ can be calculated using

$$r_{DA} = \frac{F_D}{F_A} \frac{\Phi_A \varepsilon_A}{\Phi_D \varepsilon_D} \qquad (4)$$

where $F_A$ is the emission of acceptor in the mixed FRET sample, upon excitation at the acceptor maximum. The corrected FRET efficiency $E_{cor}$ is calculated from $E_{app}$ and $r_{DA}$

$$E_{cor} = E_{app} \frac{1 + r_{DA}}{r_{DA}} \qquad (5)$$

as derived by Bykova and Zheng[64], assuming random association of donor- and acceptor-labelled NTS1. This correction accounts for the dimerisation of like fluorophore-tagged receptors (i.e., NTS1-A488-NTS1-A488 and NTS1-A555-NTS1-A555 dimers), which leads to an underestimation of the FRET efficiency. The corrected FRET efficiencies from multiple FRET samples were then averaged, and subjected to statistical analysis using R (RStudio, Inc.) using a Bayesian alternative to the two-sample $t$-test, BEST[65].

**DEER**. NTS1 proteoliposomes (containing 5–10 nmol receptor) were pelleted (100,000 g, 3 h, 4 °C) and resuspended in ~10 µL of detergent-free buffer (50 mM Tris-HCl pH 7.4, 50 mM NaCl, 1 mM EDTA, 30% (v/v) deuterated glycerol), yielding a final concentration of ~100–200 µM. Samples were loaded into 1.6 mm (outer diameter) quartz tubes (Wilmad-LabGlass) and flash-frozen in liquid nitrogen. The background dimensionality was determined to be 2.3 from control experiments with MTSL-labelled NTS1 (S172C) reconstituted together with unlabelled receptor at a 1:3 molar ratio. DEER traces (using 3- and 4-pulse PELDOR sequences) were recorded at 50 K at Q-band on an ELEXYS E580 equipped a SuperQ-FT bridge (Bruker) with 2 mm split-ring resonator (EN-5107D2, Bruker). Resulting 3- and 4-pulse DEER traces were phase-corrected using DeerAnalysis 2013[66], and then stitched together using MATLAB 2013 (MathWorks) by least-squares fitting as per the DEER-Stitch method[67,68]. Distance distributions were derived from stitched data using DeerAnalysis 2013.

**Modelling of dimer interface**. A wild-type (wt-)NTS1 model was built from the crystal structure of the NTS1 thermostabilised mutant (PDB code 4BUO, chain B[26]) by back-mutating it to its native sequence, followed by minimisation and optimisation of the hydrogen bond network around 4 Å of the back-mutated residues, using MOE[69]. Putative parallel GPCR dimerisation interfaces were identified from a thorough analysis of all GPCR crystal structures deposited in the Protein Data Bank to date; structures of the ß1 adrenergic (ß1AR, PDB 4GPO[32]), CXC chemokine type 4 (CXCR4, PDB 3OEO[35]), histamine 1 (H1R, PDB 3RZE[34]), κ-opioid (κOR, PDB 4DJH[36]) and µ-opioid (µOR, PDB 4DKL[33]) receptors were employed as templates for the construction of structural NTS1 dimer models in combination with the wt-NTS1 model. The specified NTS1 monomer crystal structure (PBD 4BUO[26]) was prioritised as a modelling template because it does not contain a fusion protein included in other crystallised NTS1 constructs (PDB 4GRV, 4XEE, 4XES, 5T04)[25,70,71], has an intact helix 8, and was solved to higher resolution than other thermostabilised NTS1 structures lacking fusion proteins (PDB 3ZEV, 4BVO, 4BWB)[26]. Seven different structural dimer models were constructed using Modeller[72], representing four different classes of dimerisation interfaces comprising of (1) TM1-2-H8 (ß1AR, κOR and µOR); (2) TM3-4 (H1R); (3) TM3-4-5 (CXCR4 and ß1AR); and (4) TM5-6 (µOR)[32–36]. The best DOPEHR-scored[73] solution of each model was used for subsequent MD simulations.

**Molecular dynamics simulations**. Using the CHARMM-GUI Martini webserver[74], the seven dimer models were converted to coarse grained (CG) representation and embedded in a BPL mimicking membrane composed of POPC:

POPS:POPE:cholesterol (15:22:39:24), and solvated in the presence of 150 mM NaCl. The force field Martini 2.2[75] was employed and, after careful equilibration, 10 μs of production CG MD trajectory was reached. The first microsecond of simulation was considered part of the equilibration so it was not considered for analysis. The protein CG beads of the remaining 9 μs were converted to atomistic (AT) representation[40] at 1 ns intervals, giving rise to 9000 frames of AT trajectory.

**Dimer interface analysis.** MD-derived models were compared to the experimental ensemble FRET data. For each of the converted CG-to-AT frames, the intradimer $C_ß–C_ß$ distances between carbons at the positions labelled in the FRET experiments (of alike transmembrane domains, i.e., TM1–TM1, TM2–TM2, etc.) were measured. These MD-derived $C_ß–C_ß$ distances ($R$) were converted to FRET efficiencies ($E_{FRET}$) using the relation $E_{FRET} = ((R_0^6)/(R_0^6 + R^6))$, and the literature value of $R_0 = 7.0$ nm for the Förster radius of the A488–A555 donor and acceptor pair[31].

These estimated FRET efficiencies for all label sites for each model were then compared to the experimental ensemble FRET efficiencies; the resulting average FRET efficiency from all 127 possible linear combinations of these models, varying the number of included models from one to seven, were tested against the experimental data. Specifically, a vector **M** with eight elements, where each element represents the estimated intradimer FRET efficiency for each of the labelling positions, was created for each linear combination of models by (element-wise) summation of the vectors for the individual models, $M_i$, multiplied by a coefficient, $c_i$, which gives the fractional contribution of the model, i.e., $\sum_i c_i M_i$, where $i$ varies from one to seven (representing the seven different dimer interface models built) and $\sum_i c_i = 1$. Using the quadprog optimisation package for R in RStudio, the optimal values for the coefficients $c_i$ were determined to minimise the residual sum of squares (RSS) between the elements of **M** and the equivalent eight-element vector containing the experimental FRET efficiencies. The Akaike information criterion (AIC) value for each model fit was used to assess the relative likelihood of the different models. An *F*-test was used to test the statistical significance of the difference between fits of the FRET data collected in the absence or presence of agonist. The combinatorial models with the lowest RSS and AICs were then selected for further analysis.

Theoretical DEER distance distributions were generated using MMM (with the MTSL rotamer library at 175 K)[76] for the NTS1 dimer models with dimer interfaces TM1–2–H8 based on ß1AR, TM3–4 based on H1R, and TM5–6 based on μOR for ten snapshots of the MD simulations for each model, which were then averaged for each model. Using the same approach as for the bulk FRET data, the calculated distance distributions for the three models and their linear combinations were fitted to the experimental DEER data, where the AIC values were used to assess the relative likelihood of the different models.

**Data availability.** Data and code supporting the findings of this manuscript are available from the corresponding authors upon reasonable request.

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

## Acknowledgements

We thank the Centre of Advanced ESR (CAESR, University of Oxford) for access to EPR equipment, and Jeffrey Harmer and William Myers for technical support. We thank Mark Newton and Christopher Wedge (Physics Department, University of Warwick) for access to Q-band EPR equipment and technical assistance during preliminary measurements. We also thank Gareth Jones (Velindre Cancer Centre) for guidance in the approach to MC simulation, and Stijn van Weezel (School of Economics, University College Dublin) for assistance with the statistical analysis. We thank Huanting Liu and Jim Naismith (University of St. Andrews) for the donation of the TEV-His[6] construct. We are grateful to GLISTEN (COST Action CM1207). This work was supported by the BBSRC (BB/G019738/1 to MIW and AW) and EPSRC (EP/D048559/1, to AW as co-PI). O.K.C. is supported by a Cardiff University SBP Research Fellowship. P.M.D. was supported by MRC (G0900076/1 to AW).

## Author contributions

O.K.C. and A.D.G. established the protocol for the smFRET experiments, which were conducted and analysed by O.K.C. O.K.C. built the microscopy setup and developed and wrote the single-molecule analysis code. A.D.G. established the protocol for the ensemble FRET experiments. P.M.D. produced all protein samples, and conducted and analysed the ensemble FRET and DEER experiments. J.C.M.G. and C.d.G. built homology dimer models and J.C.M.G. performed MD simulations. P.M.D. performed the dimer model analysis. O.K.C. designed and conducted the MC simulations. P.M.D., A.D.G. and O.K.C. wrote the manuscript, with input from the other authors. A.W. and M.I.W. supervised the project and secured the funding.

## Additional information

**Competing interests:** The authors declare no competing interests.

