## [Peer Review File(PDF 400 kb) · Nature Communications]

Reviewers' comments:

Reviewer #1 (Remarks to the Author):

I enjoyed this manuscript. It addresses a question of interest and presents noteworthy results. As the authors note, the biological importance of homodimerization at the neurotensin receptor (which is studied here) is unclear, but there's been a good deal of debate about how class A GPCRs dimerize, and this manuscript helps answer those questions.

At a technical level, I'm generally satisfied with the manuscript. One concern is that in the "Dimerisation Interface" section, when the authors point out that the experimental data (FRET and DEER) fits models with more than one interface better than models with one interface, I think they need to quantify the statistical significance of that conclusion and carefully explain how they did so. This is among the most important conclusions of the paper, and I'm having trouble assessing confidence level from Fig. S4.

Minor comments:

* The assumption of a symmetric dimer seems central to much of the analysis here. Could you defend it better, or qualify your conclusions?

* p. 12: "To our knowledge, here, we demonstrate the first evidence for a GPCR dimer "rolling interface" in which monomers sample a number of different relative conformations." Do you mean that monomers sample different conformations, or dimers? Also, is this really the first evidence for that? I've seen similar arguments in papers from Shi & Javitch.

Reviewer #2 (Remarks to the Author):

The oligomerisation state of GPCRs is an important and unresolved problem. In this paper NST1 is studied as a model GPCR in bilayers in detail using single molecule FRET based methods. The findings are potentially interesting and important to the field suggesting that dynamic dimers are formed for this particular GPCR and that multiple dimer conformations can be formed. However there are a number of key control experiments that need to be performed to confirm the current interpretation of the data. This is particularly important to properly resolve this ongoing controversy. At present there are other possible interpretations of the data and these need to be ruled out.

1. It is essential to perform a control of two non-interacting proteins labelled with Cy3 and Cy5 and show that at comparable densities no " dimers" are detected or determine what fraction of the observed events are due to random interactions and take account of this in the analysis. In my view simulations are not sufficient to address this key point. Related to this a positive control where the donor and acceptor are on the same protein would also be needed to show the expected behaviour of a dimer, if formed.

2. What is the evidence that the GPCR is still functional when labelled? This is also a key experiment that needs to be done since if the protein is no longer functional and does not signal then the results are of no relevance. How was the position for the label decided ?

3. The dynamics of high and low FRET states could be purely dye photophysics. It would be essential to perform experiments with a different FRET pair and show that the same results are obtained. A different FRET pair is used for the bulk FRET studies than the single molecule data so this should be straightforward

4. The bulk FRET should also be performed with a second FRET pair to confirm the results and rule out any effects due to dye photophysics.

5. The abstract and discussion assume that the results are applicable to all GPCRs while they only hold for this specific GPCR in bilayers. Dynamics is likely to be altered in the cell membrane. It is also not clear that comparison to previous data obtained for muscarinic receptors on cells, for example, is justifiable. I think this point needs clarification and the authors need to justify the assumption that what is measured for NTS1 is general or tone this down.

6. I found it hard to work out how many independent experiments had been performed for the experiments presented and the statistics was not clear in a number of places. For example on page 4 what is the measured diffusion coefficient for the monomer and dimer and did the authors observe a statistically significant difference in diffusion constant as they claim? On this point there is only a 2-fold increase in particle radius in one direction so what do they predict the change in diffusion coefficient to be and do the results agree?

7. Were any single molecule experiments performed in the presence of agonist and what was the outcome?

Overall key control experiments are needed to support the authors' interpretation of their data and rule out other possible contributions. It would be a much stronger and convincing paper if the authors ruled out other possible effects that could explain their data by doing the appropriate control experiments. The field needs this level of rigour to address this important and unresolved problem.

Reviewer #3 (Remarks to the Author):

Although reported almost 20 years ago, the notion of GPCR dimers is still a matter of intense debate. Although reported using various forms of energy transfer approaches, most descriptions of GPCR dimers were done on recombinant receptors expressed in heterologous cells, and often with over-expressed receptors. The reality of this phenomenon in native tissue, and the exact role of dimerization in controlling GPCR function remains unclear. Moreover, several studies based on single molecule tracking illustrate the presence of a majority of GPCR monomers at low expression density, then close to the native expression level of these membrane proteins. In the present study, the authors used a series of state-of-the-art approaches to examine the ability of the neurotensin receptor to dimerize. It is shown that, at a physiological density, about 10% of the NTS1 receptors are transient dimers (and not larger oligomers) when reconstituted into a lipid bilayer, with a half-life of association of 1-2 seconds. Their data also convincingly illustrate the ability of this receptor to associate via multiple interfaces. Taken together, the experimental data, also supported by *in silico* studies, support the actual view of class A GPCR dimerization, and bring further details in what appears a very dynamic process. However, I am surprised that the authors do not discuss much the consequences of these observations regarding GPCR signaling. If the process described corresponds to what is really happening in real cells, I doubt such a process influences signaling, as it is too variable (the functional consequence being very likely different depending on the dimerization interface), and in my opinion too transient. However, one cannot exclude the possibility that scaffolding proteins, or local higher receptor density in specific cellular sub-compartment, leading to a higher proportion of GPCRs associated into more stable dimers.

Major points:

1 – The paper is very difficult to follow, with many key information needed to understand the data hidden in the manuscript. For example, it should be made clear that the receptors labeled with the acceptor are in excess relative to those carrying the donor. This is just an example, but nicely

illustrate why the paper is very difficult to follow.

2 - According to the approach used, the authors cannot control the orientation of the NTS1 receptor in the artificial membrane. One is then expected that half of the receptors are oriented one way (N-terminal end on one side of the bilayer) while the other half is oriented the other way (C-terminal end on the same side of the bilayer). The main issue is then to know whether such possibility influences the data reported since one cannot exclude the possibility that differently oriented proteins can associate into dimers, as already observed in some crystals. Although I agree that such dimers made of differently oriented proteins may not be detected by the technologies used (the distance between the fluorophores being too large), one cannot exclude the possibility that these "dimers" prevent, or at least influence the association of two receptors similarly oriented. I would recommend the authors to analyze the orientation of their proteins in their assay.

3 - The authors argue that multiple possible dimer interfaces exist leading then to propose a "rolling" model for dimer formation. Surprisingly, they also argue that larger oligomers do not exist, or at least cannot be detected. To me such conclusion needs more analysis. Indeed, if multiple association interfaces exist, I see no reason why larger oligomers could not form. Although the single molecule data are consistent with the absence of detectable large oligomers, such experiments are done at relatively low density. What would happen at higher densities, as reported by Calebiro et al. PNAS 2013 from the Lohse's group?

4 - If the authors' proposal is correct, then any possible allosteric interaction possibly occurring between protomers in a dimer will very likely be different depending on the interface involved. This, by itself does not support an important role for dimerization in all the possible cooperativity phenomena reported. How can the authors reconcile their data with previous studies?

5 - Many of the data shown do not satisfy the expected quality needed for a convincing story. For example, this is well illustrated by Fig. 4. Panel A: are these real data, or it is just a scheme? This is unclear, though the absence of any scales on the Y and X axes suggests a scheme rather than real data. Panel B shows variations in the acceptor emission signal. It is argued that only two FRET efficacies can be identified. But looking at the data, I feel a third intermediate state may well be defined. This is why real data, and a complete analysis of what is shown in panel A is required. Also, as they stand, the illustrated data cannot convince any one these are FRET data, rather than just variation on the acceptor emission of due to photophysics properties of signal recordings. The authors should show the changes in the donor emission, and show the intensities of both donor and acceptor emissions are anti-correlated. Estimation of the basal signal, such as that obtained after donor photobleaching should be included. In the same line, the figure legends are always insufficiently detailed for a non specialist reader to understand.

Specific points:

1 - The interpretation of the data assumes that every receptor is labeled with a fluorophore. If not, a dimer composed of a fluorescent protomer and an unlabeled protomer will appear as a monomer. Has the proportion of labeled receptors been quantified? How? And how does this quantification affect the interpretation of the data? Although the authors claimed they took this into consideration, more information on how the proportion of unlabeled receptor was quantified is needed. It is true that the unlabeled receptors are taken into account in sup methods S1, but how was this fraction determined?

2 - The authors always talk about receptor densities in the "physiological" range. Has the native NTS1 receptor density been determined? Was this density a mean density, and how does it vary depending on the cellular sub-compartment?

3 - For class C GPCR, a rolling model for 7TM association has been reported, and a change in the dimer interface has been associated with receptor activity. These papers must be discussed (Xue et al., Nat Chem Biol 2015; Kim et al., PNAS 2017).

4 - As based on the data shown in Fig 1c and d, the referee is not convinced of any significant difference between the diffusion coefficient of the dimers (acceptor diffusion) and monomers (donor diffusion).

5 - Page 7, second paragraph, please indicate the position of the labeling here so that the reader

do have to look for it.

Reviewer #4 (Remarks to the Author):

The authors studied dimerization of recombinant modified with fluorescent moieties or spin labels mutants of neurotensin receptor NTS1 in liposomes, using FRET and DEER. The authors conclude that NTS1 receptor dimerizes transiently, that the rate and extent of dimerization depend on receptor concentration in the membrane, and that multiple interfaces can be involved (rolling dimer interface model). They also found that at least one mode of dimerization is suppressed by the agonist. The authors hypothesize that dimerization can play a role in the regulation of receptor activity.

As far as the dynamics of receptor dimerization goes, the data with NTS1 support earlier findings with the M1 muscarinic (ref 28) and N-formyl peptide receptor (ref 29). Their inferences regarding the effect of dimerization on signaling are logical, but these are just inferences. In addition, several experimental, conceptual, and presentation issues are not addressed in the present version of the manuscript.

Experimental and conceptual issues:

1. The authors should present the data proving functionality of the purified labeled NTS1 constructs (at least their ability to bind neurotensin).
2. The authors should discuss possible orientation of receptors in the lipid bilayer (if it is random, the calculations are off).
3. The authors should explain how their "rolling dimer interface" model is distinguished from the model of random receptor encounters without dimer formation, which would also involve different receptor surfaces. In particular, they should demonstrate that their data are not compatible with the latter model.
4. The results of simulations should be clearly distinguished from the actual data, as simulations are based on certain assumptions, which may or may not be correct.
5. The authors do not present any data on the effect of dimerization on signaling. Their inferences, however logical, should be clearly presented as speculation.

Presentation issues:

6. In fact, not only G proteins were shown to couple to monomeric GPCRs in vitro, but also arrestins (J Mol Biol. 2010 Jun 11;399(3):501-11; J Biol Chem 2011 Jan 14;286(2):1420-8). Just like in case of G proteins (ref 53), these findings were supported by the crystal structure of the arrestin-receptor complex (Nature. 2015 Jul 30;523(7562):561-7).
7. The authors should clearly state that many GPCR crystal dimers are in a definitely non-physiological anti-parallel orientation. This suggests that one should interpret all dimers found in crystals with caution.
8. Some editing is needed. There is a typo in ref 72; throughout the text, please correct verbs, as "data" is a plural noun, datum is a corresponding singular; etc.

Reviewer #1 (Remarks to the Author):

I enjoyed this manuscript. It addresses a question of interest and presents noteworthy results. As the authors note, the biological importance of homodimerization at the neurotensin receptor (which is studied here) is unclear, but there's been a good deal of debate about how class A GPCRs dimerize, and this manuscript helps answer those questions.

1.1 At a technical level, I'm generally satisfied with the manuscript. One concern is that in the "Dimerisation Interface" section, when the authors point out that the experimental data (FRET and DEER) fits models with more than one interface better than models with one interface, I think they need to quantify the statistical significance of that conclusion and carefully explain how they did so. This is among the most important conclusions of the paper, and I'm having trouble assessing confidence level from Fig. S4.

Reply: The conclusion that NTS1 can form dimers via multiple interfaces rather than a single interface comes from collective observations by smFRET, bulk FRET, and DEER experiments. Firstly, in the smFRET data, the acceptor fluorescence intensity distribution (Fig. 2b) is better fit by two log-normal components (representing both high and lower efficiency energy transfer) than by a single log-normal distribution (see new Fig. S6 and S7), which is also supported by the distribution of the observed smFRET efficiencies (see new Fig. 4c). Given this observation, we attempted to fit the bulk FRET data by a combination of dimer interface models (Fig. S12), and the relative quality of the different model fits was initially assessed from the fit residuals, while the models that best fit the FRET data were subsequently only qualitatively compared to the DEER data (Fig. 5). We have now included the Akaike information criterion (AIC) value for each model fit to assess the relative likelihood of the different combinatorial models, both for the bulk FRET data (Table S3), and the DEER data (Table S4). We have also assessed the relative likelihood of the Gaussian mixture model fits to the single-molecule intensity distributions (Fig. S6); specifically to account for differences in sample size between the intensity distributions for the three samples (Fig. S6a-c), we bootstrapped the data taking 1000 random samples of $n=1000$ for each of the three samples, fitted a one- to four-component Gaussian model to the data, and calculated the Bayesian information criterion (BIC) value to assess the relative likelihood of each of the models. The resulting BIC values are now summarised in the box plots shown in Fig. S6g-i.

→ We agree with the reviewer that assessing the relative quality of the different model fits is important and we have adjusted the relevant sections in the manuscript text, and methods section, and have added additional supplementary information (Table S3 and S4, and Fig. S6 and S7). Furthermore, we have attempted to better highlight the fact that our conclusion of multiple dimerisation interfaces stems from a combination of techniques rather than a single experiment, which also bolsters our confidence in this conclusion.

Minor comments:

1.2 The assumption of a symmetric dimer seems central to much of the analysis here. Could you defend it better, or qualify your conclusions?

Reply: The observation of multiple dimerisation interfaces in the smFRET experiments does not rely upon the assumption of a symmetric dimer, nor does it preclude the presence of asymmetric dimerisation interfaces. Nevertheless, to subsequently explore possible dimerisation interfaces that might fit the bulk FRET data, in the absence of other high-resolution information on GPCR dimers, we used the available crystallographic "dimer"

interfaces, as these are likely to represent stable, energetically favourable interfaces (see also point 4.7). These are indeed symmetric dimers, and interestingly, a crude docking approach based on surface complementarity (Patchdock, see also Fig. S12) also produced a symmetric dimer. However, we do not exclude the possibility of asymmetric dimer interfaces, we just did not have adequate models for such dimers, and the dimer interface models we propose represent just one possible explanation of our data.

→ We have adjusted the discussion of the dimer interface models as per the reviewer's suggestion.

1.3 * p. 12: "To our knowledge, here, we demonstrate the first evidence for a GPCR dimer "rolling interface" in which monomers sample a number of different relative conformations." Do you mean that monomers sample different conformations, or dimers? Also, is this really the first evidence for that? I've seen similar arguments in papers from Shi & Javitch.

Reply: With the "rolling interface" model we propose that the dimerisation interface of NTS1 dimers is variable, i.e. a dimer samples different configurations, by rotation of the monomers (around their axis perpendicular to the membrane) or "rolling" relative to each other in the dimer within the dimer lifetime. Indeed, the possibility of multiple dimerisation interfaces has been previously proposed based on evidence from mainly cross-linking (e.g. Guo et al. 2008, Xue et al. 2015, and Hu et al. 2013¹⁻³) and computational studies (e.g. Johnston et al 2011, Pluhackova et al. 2016, Kim et al. 2017), as well as an ensemble FRET study by Marsango et al. 2015. However, in our view, with the data presented here, we extend the model from the possibility of the formation of dimers in different configurations that do not interconvert directly, to the formation of a metastable dimer, which samples multiple interfaces with varying stability on a per molecule basis. Or in other words: our "rolling dimer" interface differs in that it is not different dimers that are present in different configurations (i.e. having different dimer interfaces), but discreet dimers that sample many states within the dimer lifetime. Our evidence for the presence of multiple interfaces relies on the combination of both the smFRET and the ensemble FRET and DEER experiments, and is indeed in line with previously published work on other GPCRs, while the "rolling dimer" is evidenced by the fact that we do not see individual dimer trajectories falling into the distinct low- and high-FRET populations that we observe on a spot-by-spot basis (Fig. 2b) - by way of control we establish that FRET capable monomeric protein, produces just a single state consistent with a fixed TM-TM distance (Fig. 2c and S6). Rather, in the dimer, both high- and low-FRET states, corresponding to average TM4-4 distances of ~1.5 and ~5 nm are observed within individual dimer trajectories (Fig. 4d and S8), where such large distance changes are suggestive of dynamic dimer reconfiguration. This is further characterised by the threshold crossing analysis presented in Fig. 4e and discussed in more detail in Supplementary Methods S6.

→ We have expanded and clarified our discussion of the novelty of our findings and have added the references suggested by reviewer #3 (point 3.10) to this discussion.

Reviewer #2 (Remarks to the Author):

The oligomerisation state of GPCRs is an important and unresolved problem. In this paper NST1 is studied as a model GPCR in bilayers in detail using single molecule FRET based methods. The findings are potentially interesting and important to the field suggesting that dynamic dimers are formed for this particular GPCR and that multiple dimer conformations

can be formed. However there are a number of key control experiments that need to be performed to confirm the current interpretation of the data. This is particularly important to properly resolve this ongoing controversy. At present there are other possible interpretations of the data and these need to be ruled out.

2.1 **(a)** It is essential to perform a control of two non-interacting proteins labelled with Cy3 and Cy5 and show that at comparable densities no "dimers" are detected or determine what fraction of the observed events are due to random interactions and take account of this in the analysis. In my view simulations are not sufficient to address this key point. **(b)** Related to this a positive control where the donor and acceptor are on the same protein would also be needed to show the expected behaviour of a dimer, if formed.

Reply: (a) We have previously demonstrated in single-colour diffraction-limited experiments in droplet interface bilayers that interacting alpha-hemolysin monomers colocalise for no longer than 5 ms where oligomerisation does not proceed to the stable heptameric state.⁴ The individual monomer-monomer interaction of these membrane proteins is short-lived, despite their ability to ultimately form stable heptameric pores. The reported experiments find that heptamerisation is rare, but occurs rapidly (<5 ms) to produce a small number of stable heptamers in a high density monomeric population, with no intermediate oligomers persisting beyond the 5 ms time scale. Consequently, this benchmark provides strong evidence that the time-scales of interaction for NTS1 monomers reported in this work (90+ ms) can reliably be attributed solely to *bona fide* and stable (although transient) protein-protein interaction and therefore dimerisation.

The contribution of chance (i.e. non-interacting), diffraction-limited co-localised trajectories to observed dimerisation events has been demonstrated to be negligible on these time scales in live cell single-molecule imaging.⁵ In the well-controlled *in vitro* DIB system reported here we can have greater confidence that measured interactions are driven by molecular interaction alone, by virtue of making measurements in a minimal system with membrane diffusion unhindered by the cytoskeleton or other cellular features. Here, we can extend the approach of Hern et al.⁵ and experimentally determine the contribution of prolonged chance co-localised events by comparing trajectories in the continuously visualised Cy3 channel in multiple experiments. To this end we combine all measured Cy3-NTS1 trajectories from five separate experimental videos to provide an equivalent trajectory density to that of the non-visualised Cy5-labelled species. Every trajectory was then tested for spatial and temporal coincidence with all trajectories in a sixth Cy3-NTS1 video, serving as a set of donor trajectories. Coincidence was determined at different spatial proximities determining the number of chance co-localisations over different time-scales and spatial distances. At a co-incident diameter of 200 nm, approximating the proximity limit for determining diffraction-limited colocalisation, we find an equivalent of 15.1% of our measured dimer traces would be attributable to the chance colocalisation of two receptors within 200 nm over three or more consecutive frames. It is notable that these represent 3 frame (10.65%) and 4 frame (4.44%) events only, with no longer lasting coincident diffusion events detected (Fig. S5). This represents the false positive rate if determination of receptor-receptor interaction were made by diffraction-limited means at our reconstituted receptor density.

In the dimerisation experiments reported here, we use FRET to afford more than an order of magnitude greater spatial proximity precision in determining interaction compared to diffraction-limited colocalisation and have accepted only interactions persisting for longer than 90 ms (3 frames), providing a stringent threshold in attributing interaction for classification as NTS1 dimers. Since we employ FRET to determine receptor-receptor interaction, two receptors must be within the order of the Förster radius, in this case approximately 5 nm, to

be detected as interacting and attributed as a dimer. At this length scale no coincident trajectories persisting for more than 3 frames (the employed cut-off) are observed (Fig. S5), demonstrating that chance coincident diffusion makes a negligible contribution to the measured dimer trajectories. By plotting the relationship of coincidences persisting for a minimum of three frames against spatial radius defining coincidence, and fitting a decay curve, we estimate a maximum false positive detection rate of 0.04% in our single-molecule FRET experiments, confirming that co-diffusing, non-interacting, species do not significantly contribute to dimerisation detections in our population of 1,167 measured trajectories. These measurements are in good agreement with analytical solutions for receptor collision which indicate the rarity of co-occupation, even momentarily, of two receptors within a collision radius of each other (of comparable magnitude to the Förster radius). This is borne out by Monte Carlo simulations and the experimentally observed steep reduction in detected dimers with small reduction in receptor density (Fig. S4a), as a direct consequence of the decreasing probability of receptor collision.

(b) Regarding the positive control in which donor and acceptor are present on the same protein: this control has been performed and the data can be found in Fig. 2c,f,i. We used a double cysteine mutant of NTS1 A90C^{1.58}-T186C^{4.42} which we labelled with Cy3 and Cy5 (generating a mix of proteins labelled with either 0-2x Cy3, 0-2x Cy5, or 1x Cy3 and 1x Cy5) reconstituted into droplet interface bilayers. Intensity analysis of the acceptor fluorescence intensity of those spots corresponding to intramonomer FRET for this doubly labelled sample (Fig. 2c) showed only one population, in contrast to the two populations observed in the case of inter-monomer FRET for dimers, validating that observation. We have restructured the discussion of the results and specifically Fig. 2, and have added Fig. S6 and S7, to make this control and its importance more clear.

→ We have added the analysis quantifying the false positive dimerisation detection (point 2.1a; Supplementary Methods S2) and accompanying figure (Fig. S5) to the SI. Furthermore, we discussed this control and the positive control requested in point 2.1b (shown in Fig. 2 and further discussed in Fig. S6 and S7) in the main text.

2.2 What is the evidence that the GPCR is still functional when labelled? This is also a key experiment that needs to be done since if the protein is no longer functional and does not signal then the results are of no relevance.

Reply: We have performed and added microscale thermophoresis (MST) experiments on the cysteine-depleted background mutant showing ligand binding with equally high affinity (1 nM) as wild-type (wt-)receptor ($K_d < 10$ nM). Furthermore, the purification protocol includes a step with a neurotensin column, where receptor is bound via interaction with the ligand immobilised on resin, and eluted with NaCl, thus ensuring that the final purified receptor is able to bind ligand, and thus likely to represent properly folded, functional receptor. In addition, we have performed MST experiments on NTS1 fluorescently labelled at TM4 (T186C^{4.42}-Alexa488) and TM6 (V307C^{6.34}-Alexa488) as two representative mutants and wt-NTS1 where we titrated $G\alpha_{i1}$ which showed that the fluorescent receptor in detergent is able to bind the G protein α subunit. Incubation with GTP γ S strongly lowered the affinity, evidencing specific $G\alpha_{i1}$ binding. These combined data suggest that the introduction of cysteine mutations and fluorescent labels does not hamper the basic interactions involved in NTS1 signalling, and the receptors are thus functional.

→ MST measurements demonstrating ligand and G protein binding have been added to the SI (Fig. S2 and S3).

2.3 How was the position for the label decided ?

Reply: The labelling positions were selected by analogy to a previous DEER study on rhodopsin⁶ and verified as accessible positions on a homology model, as at the time no high-resolution structure of NTS1 was available, and we rationalised that given the high sequence analogy between class A GPCRs, these sites were likely to be accessible, and that their labelling was also likely to not perturb NTS1 function, given that their labelling was non-perturbing in rhodopsin. Subsequent investigation of published crystal structures of NTS1⁷⁻¹⁰ confirmed that the chosen sites were exposed and thus suitable for labelling.

A previous computational study on NTS1 dimers that suggested that TM4 was part of the dimerisation interface.¹¹ Thus, as stated in the main text, TM4 was chosen as the labelling site in the smFRET studies based on the expectation that the interlabel distance would be short enough to give efficient FRET.

→ A brief motivation for the choice of labelling sites has been added to the methods section.

2.4 The dynamics of high and low FRET states could be purely dye photophysics. It would be essential to perform experiments with a different FRET pair and show that the same results are obtained. A different FRET pair is used for the bulk FRET studies than the single molecule data so this should be straightforward. The bulk FRET should also be performed with a second FRET pair to confirm the results and rule out any effects due to dye photophysics.

Reply:

The distribution observed in the single-molecule fluorescence spots was not only investigated for the spots in the Cy5 channel corresponding to dimeric receptor (i.e. inter-monomer FRET) where we observed the high- and low-FRET states (Fig. 2b), but we also included two controls, namely:

- Control 1 - the intensity distribution of spots in the Cy5 channel after excitation of Cy3 for a doubly labelled monomer (discussed also in point 2.1b), where the acceptor intensity was measured originating from FRET between the Cy3-Cy5 dyes attached to the same molecule (Fig. 2c), i.e. intra-monomer FRET;
- Control 2 - the intensity distribution of spots in the Cy3 channel after direct excitation of Cy3 for singly labelled receptors corresponding to monomeric receptor (Fig. 2a).

It is apparent that the intensity distributions in the above controls, unlike the dimeric species, are very well described by a single log-normal distribution, indicating that the observed bimodal dimer FRET distribution is not a consequence of inherent imaging variation (concurrently acquired control 2), nor a result of Cy3-Cy5 FRET photophysics, since for the control experiments in a doubly labelled, FRET-capable, monomeric protein (control 1) the acceptor intensity is well described by a monomodal log-normal distribution (Fig. 2).

This was confirmed by analysis of Gaussian mixture distributions with n components fitted to the single-molecule intensity data for these three species, which is shown in more detail in Fig. S6. Correcting for differences in sample size as described in point 1.1, BIC values for each model were calculated to assess the relative likelihood of single- and multi-component models, where higher values indicate a more likely model. Whilst for both controls no improvement was seen on increasing the number of model parameters (Fig. S6h-i), the FRET acceptor intensity distribution of the dimeric species is better fit by multicomponent models (Fig. S6g, and S7).

Furthermore, we have orthogonal evidence from the ensemble (FRET and DEER) experiments that is in agreement with multiple dimer configurations, which are likely to give rise to FRET states with differing FRET efficiency.

These combined observations strongly suggest that the observed multi-state intensity histogram found for inter-monomer FRET experiment (Fig. 2b) is due to the presence of different species, rather than purely dye photophysics. Indeed, we used a combination of multiple methods (single-molecule and ensemble FRET with fluorophores, and DEER with spin labels) to mitigate the shortcomings of any single approach.

→ As per point 2.1, the manuscript the discussion of the data in Fig. 2, and specifically the performed control, has been adjusted for clarity, in addition to further changes to the text to better tie together the results from the different methods.

2.6 The abstract and discussion assume that the results are applicable to all GPCRs while they only hold for this specific GPCR in bilayers . Dynamics is likely to be altered in the cell membrane. It is also not clear that comparion to previous data obtained for muscarinic receptors on cells, for example, is justifiable. I think this point needs clarification and the authors need to justify the assumption that what is measured for NTS1 is general or tone this down.

Reply:

→ We agree with the reviewer on this point and have adjusted the discussion of our results accordingly.

2.7 **(a)** I found it hard to work out how many independent experiments had been performed for the experiments presented and the statistics was not clear in a number of places. For example on page 4 what is the measured diffusion coefficient for the monomer and dimer and did the authors observe a statistically significant difference in diffusion constant as they claim? **(b)** On this point there is only a 2-fold increase in particle radius in one direction so what do they predict the change in diffusion coefficient to be and do the results agree?

Reply: (a) We have clarified the number of independent experiments, and significance of different findings throughout the manuscript (see also point 1.1). In the case of the diffusion constants for particles in the donor and acceptor channel (Fig. 1c,d), we have moved the relevant numbers from the figure legend to the main text to aid the reader. The 95% confidence intervals for the diffusion constant of the spots in the acceptor ($[1.222, 1.317] \mu\text{m}^2\text{s}^{-1}$, $n = 1,167$ trajectories) and the donor channel ($[1.609, 1.647] \mu\text{m}^2\text{s}^{-1}$, $n = 17,853$ trajectories), are now also stated in the main text. From the non-overlapping 95% confidence intervals for these diffusion constants, we can conclude that the difference between them is significant, which was confirmed by a two-sided z-test ($z = -14.12$, $p < 10^{-5}$).

(b) To address point made by the reviewer regarding the change in particle radius: of course the Saffman-Delbrück model for lateral diffusion ($D \propto -\ln[r]$) assumes spherical particles, and thus the radius r represents an “effective” radius. Considering that for the NTS1 dimer the particle radius increases by 2 only in one direction relative to that of the NTS1 monomer, we would expect to see an increase in effective particle radius of < 2 . For example, the ratio between the effective radius of a dimeric lactose transporter, LacS, and a monomeric homologue, LacY, has been reported to be ~ 1.6 , and their diffusion constants were consistent with the Saffman-Delbrück model of diffusion.¹²

From the relation $D \propto \ln[r]$, it follows that $r_1/r_2 = \exp(D_2 - D_1)$. For the diffusion constants for particles in the donor channel, $D_{\text{donor}}(\text{monomer}) = 1.63 \mu\text{m}^2\text{s}^{-1}$ (95% CI [1.609, 1.647] $\mu\text{m}^2\text{s}^{-1}$, $n = 17,853$ trajectories), and in the acceptor channel, $D = 1.27 \mu\text{m}^2\text{s}^{-1}$ (95% CI [1.222, 1.317] $\mu\text{m}^2\text{s}^{-1}$, $n = 1,167$ trajectories), this gives $r_1/r_2 = 1.43$ (rather than the factor 2 reported in the earlier version of the manuscript - this has been corrected), which is in line with our expectations.

→ The number of experiments has been moved from the figure legend to the main text, and the discussion of the diffusion constants has been adjusted.

2.8 Were any single molecule experiments performed in the presence of agonist and what was the outcome?

Reply: These experiments were not performed to a great extent. Initial trials did not show dramatically different behaviour. Previously published bulk experiments showed no effect of NT on the extent of dimerisation.¹³ This of course does not rule out any potential effect of agonist e.g. on the high/low-FRET distribution of the acceptor intensity (or in other words, the dimer configuration) or on the dimer lifetime, that we might be able to probe by single-molecule experiments. However, from our initial trials we anticipate any changes on dimer lifetime or productivity to be small, meaning that we would need to conduct a very extensive amount of additional experiments to be able to show significant differences, given the variability of protein reconstitution between experiments (see Fig. S4), which when coupled with non-linear concentration dependency of collisions generates measured variability. This is discussed in Supplementary Methods S4 - Estimation of error. Thus, although these experiments could potentially be interesting, we feel that they would go beyond the scope of the current study.

Overall key control experiments are needed to support the authors' interpretation of their data and rule out other possible contributions. It would be a much stronger and convincing paper if the authors ruled out other possible effects that could explain their data by doing the appropriate control experiments. The field needs this level of rigour to address this important and unresolved problem.

Reply: We have conducted control experiments of donor and acceptor on the same protein which show only a single fluorescence intensity population (see point 2.1b) and used existing published experimental data, together with additional experimental analysis to establish the contribution of chance interactions towards our measured dimers (see point 2.1a). In line with others reporting colocalisation of singly labelled receptors in cells, we find a negligible probability of chance coincident localisation over multiple frames. Studies on the interaction of alpha-hemolysin monomers provide a credible frame of reference that demonstrates that the dimer interactions reported here, persisting on the time-scale of 100s of milliseconds to seconds are the result of *bona fide* protein-protein interactions. These time scales are further corroborated by dimer lifetimes reported for other GPCRs by single-colour colocalisation imaging in cells. We have now made reference to these controls and (more clearly) included statistical tests of findings. In addition, we have altered the discussion of the controls performed and the results in general, as discussed under points 2.1-2.8 and hope this satisfies the concerns of the reviewer.

Reviewer #3 (Remarks to the Author):

Although reported almost 20 years ago, the notion of GPCR dimers is still a matter of intense debate. Although reported using various forms of energy transfer approaches, most descriptions of GPCR dimers were done on recombinant receptors expressed in heterologous cells, and often with over-expressed receptors. The reality of this phenomenon in native tissue, and the exact role of dimerization in controlling GPCR function remains unclear. Moreover, several studies based on single molecule tracking illustrate the presence of a majority of GPCR monomers at low expression density, then close to the native expression level of these membrane proteins. In the present study, the authors used a series of state of the art approaches to examine the ability of the neurotensin receptor to dimerize. It is shown that, at a physiological density, about 10% of the NTS1 receptors are transient dimers (and not larger oligomers) when reconstituted into a lipid bilayer, with a half-life of association of 1-2 seconds. Their data also convincingly illustrate the ability of this receptor to associate via multiple interfaces. Taken together, the experimental data, also supported by *in silico* studies, support the actual view of class A GPCR dimerization, and bring further details in what appears a very dynamic process.

3.1 However, I am surprised that the authors do not discuss much the consequences of these observations regarding GPCR signaling. If the process described corresponds to what is really happening in real cells, I doubt such a process influences signaling, as it is too variable (the functional consequence being very likely different depending on the dimerization interface), and in my opinion too transient. However, one cannot exclude the possibility that scaffolding proteins, or local higher receptor density in specific cellular sub compartment, leading to a higher proportion of GPCRs associated into more stable dimers.

Reply: As previously argued by Bouvier and Hebert¹⁴, the observation that GPCR oligomers are more transient than, e.g. stable oligomers found for ion channels, does not preclude any role in modulation of signal transduction. Indeed, GPCRs show complex signalling patterns, activating different G proteins (NTS1 for example has been shown to activate Gq, Gi, and Gs) as well as arrestins. Such complex signalling would require significant conformational plasticity, which may be modulated by local environment, and indeed dimerisation. As previously proposed by Bouvier and Hebert¹⁴, “multiple weak interfaces allow for allostery and the requirements of distinct organisational designs as needed by the cell”, i.e. the transience and variability of the dimerisation may in fact be an important feature of its role in signal regulation, rather than a limitation. Indeed, as previously argued by Gurevich and Gurevich¹⁵, “many regulatory protein-protein interactions have nanomolar affinities and last for seconds or even less”, (e.g. receptor tyrosine kinases), thus the transient nature of GPCR dimers does not preclude any biological importance. Of course, we agree with reviewer #3 that dimerisation may be influenced by other cellular components, not included in the present study, which may modulate the lifetime and/or preferred interface of the dimers in a given cellular environment. The inherent flexibility of the dimerisation process may be important to facilitate different dimer arrangements in different environments or activation states, allowing for distinct allosteric fine tuning of signalling as required by the cell. We do note that any interpretation of the physiological consequences of our observations remains speculative at this stage, but hope these experimental observations provide a new paradigm within which researchers can study the physiological implications of GPCR dimerisation.

→ We have expanded our discussion on the transience and variability of dimerisation.

Major points:

3.2 The paper is very difficult to follow, with many key information needed to understand the data hidden in the manuscript. For example, it should be made clear that the receptors labeled with the acceptor are in excess relative to those carrying the donor. This is just an example, but nicely illustrate why the paper is very difficult to follow.

Reply:

→ We appreciate the concern of the reviewer and have altered and restructured the manuscript accordingly to improve clarity.

Regarding the ratio of donor and acceptor labelled species, we appreciate the value of this information and have now stated this in the main manuscript body in addition to the detail in the supplementary materials. We have also added the rationale behind this to the methods - "This concentration ratio ensured a greater prevalence of FRET capable dimerisation compared to equimolar labelling conditions, affording the measurement of an increased number of dimers, whilst retaining single-molecule resolution in the donor channel."

3.3 According to the approach used, the authors cannot control the orientation of the NTS1 receptor in the artificial membrane. One is then expected that half of the receptors are oriented one way (N-terminal end on one side of the bilayer) while the other half is oriented the other way (C-terminal end on the same side of the bilayer). The main issue is then to know whether such possibility influences the data reported since one cannot exclude the possibility that differently oriented proteins can associate into dimers, as already observed in some crystals. Although I agree that such dimers made of differently oriented proteins may not be detected by the technologies used (the distance between the fluorophores being too large), one cannot exclude the possibility that these "dimers" prevent, or at least influence the association of two receptors similarly oriented. I would recommend the authors to analyze the orientation of their proteins in their assay.

Reply: The reviewer is correct in stating that we do not have experimental control over the reconstitution orientation of NTS1 in the artificial membrane. NTS1 reconstitution in liposomes using the same methods as used in this study has previously been reported by our laboratory (Harding et al. 2009) to yield symmetric reconstitution with ~50/50 distribution of both orientations.¹³ The reconstitution orientation in the droplet interface bilayers (DIBs) used in our single-molecule work is unknown. This leads to a number of theoretically possible scenarios:

- 1) Reconstitution is asymmetric (i.e. all monomers are inserted in the same orientation);
- 2) Reconstitution is symmetric (i.e. monomers are inserted in random orientations), and anti-parallel dimers do not form;
- 3) Reconstitution is symmetric, and anti-parallel dimers can form.

In scenario #1 our analysis would not be affected.

In scenario #2, our reconstitution efficiency would effectively be off by a factor 2, since each receptor will only be able to interact with half of the receptors present. This would mean that our measured/derived parameters would (only) be underestimated by a factor of 2, specifically: the dimer half-life, $t_{1/2}=1.2$ s, (95% CI [0.65,1.54] s), and the $K_d=7.1$ μm^2 (95% CI [6.7,12.3] μm^2). The rate of dimer formation, $k_{on}=0.081$ $\mu\text{m}^{-2}\text{s}^{-1}$ (95% CI [0.072,0.088] $\mu\text{m}^{-2}\text{s}^{-1}$) is measured directly from the experiment (as detailed in Supplementary Methods S1) and would not be affected.

In scenario #3, in theory, any measured parameter would represent a population-weighted average of the parallel and the non-physiological anti-parallel dimers. However, we would not expect to observe a large part of any hypothetical anti-parallel dimers via FRET because, as the reviewer notes, the inter-label distance would be too large to give rise to efficient energy transfer for most hypothetical anti-parallel dimer configurations.

Whilst, we cannot definitively exclude any of the three aforementioned scenarios, the observed dimer half-life (1.2 s) is consistent with previous reports on other GPCR dimers (~0.1-5 s).^{5,16,17}

Additionally, the observed TM4-4 distance of ~5.0 nm for the low FRET state and shorter (~1.5 nm) for the high-FRET state are too short to both originate from anti-parallel dimers, for which we would expect a minimal distance of ~4-7 nm. Thus, neither the high-FRET state observed in smFRET, nor the short distances observed by DEER, could originate from an anti-parallel dimer. Furthermore, the smFRET threshold crossing analysis (presented in Fig. 4) showed that individual dimers sample both the low- and high-FRET states, i.e. show interconversion between these states. Both these states must thus originate from a parallel dimer to be able to interconvert, validating our conclusion that the physiologically relevant dimer samples multiple interfaces.

In addition, the relative proportion of low- and high-FRET states identified from the FRET efficiency histogram (80/20, Fig. 4c) is very similar to that identified from the dimer acceptor fluorescence intensity distribution (84/16, Fig. 2a). Notably, FRET efficiencies were only calculated from a subset of long-lived dimers (to be able to extract meaningful acceptor trajectories) and are thus more likely to include contributions from the high-FRET state. Thus, the FRET efficiency histogram is biased towards trajectories containing transitions between low- and high-FRET states, which, as argued above, cannot originate from anti-parallel dimers. The dimer acceptor fluorescence intensity histogram does not have this bias and also includes shorter traces that are more likely to only sample the low FRET states. If hypothetical anti-parallel dimers with long, but observable interlabel distances were present in significant proportions, this would be reflected in a much higher low FRET population in the acceptor fluorescence intensity distribution histogram compared to the FRET efficiency histogram, which we do not observe.

Taken collectively, we consider it reasonable to conclude that any effect must be minimal, (as is the case for example with Cy3-unlabelled dimers and Cy3-Cy3 dimers that we show are insignificant amongst the visualised donor labelled receptors).

In conclusion, while we cannot exclude the possibility of random insertion in the bilayer, and the formation of anti-parallel dimers with the data at hand, we expect only a modest effect on our results.

→ We have added a paragraph regarding the expected effects of insertion orientation in the bilayer to our discussion.

3.4 The authors argue that multiple possible dimer interfaces exist leading then to propose a "rolling" model for dimer formation. Surprisingly, they also argue that larger oligomers do not exist, or at least cannot be detected. To me such conclusion needs more analysis. Indeed, if multiple association interfaces exist, I see no reason why larger oligomers could not form. Although the single molecule data are consistent with the absence of detectable large oligomers, such experiments are done at relatively low density. What would happen at higher densities, as reported by Calebiro et al. PNAS 2013 from the Lohse's group?

Reply: Indeed, reviewer #3 is correct in that we cannot rule out that higher order oligomers can be formed *per se*. In the single-molecule FRET experiments, we only observe single-step photobleaching (Fig. 2g-i), suggesting no higher order oligomers are present at low receptor densities (~ 0.4 copies/ μm^2). Indeed, their formation in these experiments would be unlikely, given the low frequency of monomer collisions and short dimer lifetime. At higher receptor densities ($\sim 1\text{-}3 \times 10^3$ copies/ μm^2), it has previously been shown by our laboratory, that NTS1 reconstituted into liposomes using the same protocols as used here in the bulk experiments forms dimers (with roughly 90% of the receptor being present as dimers).¹⁵ Specifically, Harding et al. performed bulk FRET experiments on NTS1 reconstituted into BPL liposomes, varying the donor/acceptor ratio and fitted the resulting FRET efficiency vs. donor/acceptor ratio curve to a dimer/trimer/tetramer model, and showed that the data were consistent with a dimer model and the not with the formation of higher order oligomers.¹⁵ Experiments with spin-labelled lipids with NTS1 reconstituted into liposomes at even higher receptor densities ($>10^4$ copies/ μm^2) did suggest the presence of higher order oligomers.¹⁸ It is worth noting that these densities are significantly higher than those reported for NTS1 *in vivo* ($\sim 2\text{-}40$ copies/ μm^2)¹⁹⁻²², and those used in the study by Calebiro et al ($\sim 0.2\text{-}0.4$ copies/ μm^2).¹⁷ Thus, higher order NTS1 oligomers may be possible at very high receptor density, but we have no evidence to support their presence at the receptor densities used in our experiments, and consequently we can interpret our present data within the framework of a monomer-dimer equilibrium.

→ The discussion has been adjusted.

3.5 If the authors' proposal is correct, then any possible allosteric interaction possibly occurring between protomers in a dimer will very likely be different depending on the interface involved. This, by itself does not support an important role for dimerization in all the possible cooperativity phenomena reported. How can the authors reconcile their data with previous studies?

Reply: As for comment 3.1, we (and e.g. Bouvier and Hebert, J Physiol 592.12, 2014, p2447)¹⁴ would argue that the flexibility in the possible dimer interfaces may in fact be important for GPCRs which signal through a wide array of pathways. Previous studies on allosteric behaviour in cells show results of ensemble populations such that it is impossible to distinguish whether different dimer configurations within a population are having an effect. In particular cellular environments, certain dimer interfaces may be preferred, either through modulation by the lipid environment (e.g. see simulations by Pluhackova et al.²³) or interactions with other cellular components, which may give rise to varying allosteric functions of the dimer interface. Indeed, we argue that for example an interface made by TM5-6 may limit basal G protein coupling activity of the receptor, while the other interfaces proposed would support G protein activity.

→ We have expanded our discussion on the transience and variability of dimerisation.

3.6 Many of the data shown do not satisfy the expected quality needed for a convincing story. For example, this is well illustrated by Fig. 4. Panel A: are these real data, or it is just a scheme? This is unclear, though the absence of any scales on the Y and X axes suggests a scheme rather than real data. Panel B shows variations in the acceptor emission signal. It is argue that only two FRET efficacies can be identified. But looking at the data, I feel a third intermediate state may well be defined. This is why real data, and a complete analysis of what is shown in panel A is required.

Reply: Fig. 4a (in the previous version of the manuscript) was a reproduction of the data presented in Fig. 2b. This panel was included in Fig. 4 to indicate what the “low-FRET” and “high-FRET” populations are, and the threshold used in the threshold crossing analysis displayed in Fig. 4. For context; we originally aimed to simply the histogram plot to emphasise how the high- and low-intensity states and ascribed threshold mapped between the trajectory intensity traces and the overall histogram of acceptor intensity traces. However, we appreciate the reviewer’s concern and to avoid any confusion for the reader we have altered the figure, its legend, and the discussion of this analysis in the text.

Regarding the number of states that exist; as discussed in point 1.1 and 2.4, we have now included a more detailed description of the (statistical) analysis of the single-molecule fluorescence intensity distributions (Fig. S6). We have acknowledged that multiple FRET efficiencies exist, however, given the distribution of measured efficiencies we have been reluctant to over-interpret or over-fit multiple states to our data. Instead, we have focussed on the clearly identifiable high- and lower-FRET efficiencies in the acceptor intensity distribution. We acknowledge that the high-FRET state is likely due to a single conformation with a relatively short intradimer TM4-4 distance and that the lower FRET state could be the result of one, or more, configurations with longer TM4-4 separation. Whilst it is tempting to fit further intermediate states to trajectory traces, the rapid time scale of fluctuation, on the order of the frame time, makes this challenging. Instead we have focussed our further analysis on the transitioning between the much more readily defined high- and lower-FRET states. Indeed, the three-component model was not distinguishable from the two-component model based on the BIC values of the fits (Fig. S6g and S7). We also note for clarity that the defined threshold ascribed, does not itself represent a state but a threshold intensity above which we can be confident of dimers belonging to the high-FRET population (Supplementary Methods S6).

→ The figures, and in particular Fig. 4, and the discussion of the figures in the text have been altered to improve clarity. Probability analysis of Gaussian mixture models to the fluorescence intensity data is now explicitly included (Fig. 6) and discussed in more detail.

3.7 Also, as they stand, the illustrated data cannot convince any one these are FRET data, rather than just variation on the acceptor emission of due to photophysics properties of signal recordings. The authors should show the changes in the donor emission, and show the intensities of both donor and acceptor emissions are anti-correlated. Estimation of the basal signal, such as that obtained after donor photobleaching should be included. In the same line, the figure legends are always insufficiently detailed for a non specialist reader to understand.

Reply: We have created two-colour, registered, overlaid images to more clearly illustrate the visualisation of monomeric protein (donor, green) and dimeric protein (acceptor, red) in our single-molecule experiments. This has been added to the manuscript as an additional panel in Fig. 1 illustrating single monomer and dimer observations in addition to dimer formation (the initiation of FRET, Fig. 1e) and the termination of FRET in a dimer trajectory (Fig. 1f), where signal transitions from donor to acceptor and acceptor to donor, respectively. We have also included the corresponding videos as supplementary material. These data demonstrate that we are observing FRET and not artefactual detections. We note that the number of acceptor observations are non-linearly linked to receptor concentration, as a consequence of the reduced probability of receptor collisions as concentration decreases. During the course of our experimental work we found that dimer observations all but disappear (in our time-limited observation window) at receptor densities below about $0.2 \mu\text{m}^{-2}$, in keeping with density-dependent collisions being required for dimer formation and FRET observation (this

relationship is discussed in Supplementary Methods S4 - Estimation of error). This is consistent with observing FRET as a direct result of receptor interaction, which as our MC simulations illustrate, drops precipitously as receptor density decreases.

In the first instance we further expand on our response to point 2.4. The measured bimodal distribution of dimer acceptor intensities is observed in contrast to measurements of a double (Cy3-Cy5) labelled, FRET-capable, single receptor species, where acceptor emission does not display bimodal behaviour, but is instead well described by a single log-normal distribution (Fig. 2c). Similarly, we do not observe such a distribution for the fluorescence intensity of the monomeric Cy3 labelled protein (Fig. 2a), which is acquired simultaneously to the dimer acceptor data, demonstrating that this bimodal distribution is not a consequence of our acquisition or experimental variation, e.g. laser fluctuation.

We have included statistical information on the quality of single- and multiple-component fits to the monomer, dimer and double-label species (Fig. S6). Thus we establish with confidence that the dimer acceptor emission is described by a multimodal distribution, and is the result of at least two states: a lower proportion high-FRET component and one or more configurations giving rise to the predominant intermediate FRET state.

Furthermore, we have orthogonal evidence from the ensemble (FRET and DEER) experiments, which is consistent with multiple dimer configurations, which are likely to give rise to FRET states with differing FRET efficiency. These combined observations strongly suggest that the observed multi-state intensity histogram found for inter-monomer FRET experiment (Fig. 2b) is due to the presence of different species, rather than a photophysical manifestation. Indeed, we used a combination of multiple methods (single-molecule and ensemble FRET with fluorophores, DEER with spin labels) to mitigate the shortcomings of any single approach, and observe consistent data across these methodologies.

In summary, we have evidence from multiple techniques that is consistent with multiple dimer configurations. At the single-molecule level we observe dimer acceptor intensities consistent with at least two states, giving rise to high, and intermediate acceptor intensities.

Interestingly, despite being able to observe two intensity states within the whole population (that we do not observe in our FRET control of double-labelled protein), we are unable to classify whole dimer trajectories as belonging to one or the other of these states. We instead find that this bimodal distribution arises as a result of rapid dynamic changes within dimer trajectories (Fig. 4d and S8).

To further interrogate this, we have added further data analysis to Fig. 4 and the SI (Fig. S9) by plotting donor and acceptor intensity, extracted from the registered donor and acceptor images, as suggested by the reviewer. These data are in keeping with the observation of dynamic changes on rapid time scales, rather than discrete steps between well separated states. Further, we have include Supplementary Movie S5, depicting single-molecule FRET overlaid donor (green) and acceptor (red) illustrating donor and acceptor intensity fluctuation within a dimeric FRET trajectory. This is also illustrated by the addition of an image sequence incorporated in Fig. 4a.

We calculate FRET efficiency (E) at the single-molecule level (Fig. 4c). A histogram of single-molecule FRET efficiencies is consistent with a broad distribution of FRET efficiencies, including a broad and predominant intermediate FRET state and a smaller, but significant high-FRET population. Again, this is observed to arise as a result of variation occurring within trajectories. Calculation of average TM4-4 distance from this single-molecule FRET efficiency data is in very close agreement with that by other methods at approximately 5 nm

(Supplementary Methods S5 - Estimation of smFRET distance measurements), as is the relative proportion of high- and low-FRET conformations (80/20 by smFRET E vs. 84/16 by the acceptor intensity distribution analysis).

To avoid compounding single-molecule noise, we concentrated our analysis in understanding the dynamics of this behaviour on the acceptor intensity transitioning between the readily defined high- and lower-FRET states, rather than speculate as to the number of intermediate states comprising the intermediate FRET efficiency (as discussed in the response to 3.6 above).

We have discussed the contribution of basal signal in Supplementary Methods S4 - Estimation of error. The use of narrow band emission filters, meant that we were unable to spot detect and Gaussian fit to extract intensity on emission resulting from bleed or cross-talk. Specifically, after donor photobleaching we are unable to detect a spot in either donor or acceptor channel to extract an intensity, thus we conclude any basal contribution is minimal. We have extracted spot intensity by pixel averaging in the single-molecule intensity traces in Fig. 2 to circumvent this and this data gives an indication of basal signal and is consistent with the above. Our two-colour, registered, overlaid images (Fig. 1e-f) and Movies S1-3 and S5 also serve to illustrate this visually.

→ We have extended our data analysis (presented in Fig. 4, S6-9) and have sought to improve clarity of information in figure legends throughout the manuscript, whilst being mindful of the need to be concise.

Specific points:

3.8 The interpretation of the data assumes that every receptor is labeled with a fluorophore. If not, a dimer composed of a fluorescent protomer and an unlabeled protomer will appear as a monomer. Has the proportion of labeled receptors been quantified? How? And how does this quantification affect the interpretation of the data? Although the authors claimed they took this into consideration, more information on how the proportion of unlabeled receptor was quantified is needed. It is true that the unlabeled receptors are taken into account in sup methods S1, but how was this fraction determined?

Reply: Indeed, the labelling efficiency of the sample was taken into account in the analysis of the data (indeed as discussed in Supplementary methods S1). The average labelling efficiency of each sample was determined by comparing the absorption at 280 nm with that of the fluorophore maximum, correcting for the contribution of the fluorophore at 280 nm:

$$E_{\text{labelling}} = \frac{[\text{Label}]}{[\text{NTS1}]} = \frac{A_{\text{Label}} / \epsilon_{\text{Label}}}{(A_{280} - CF \times A_{\text{Label}}) / \epsilon_{\text{NTS1}}}$$

where A_{Label} and A_{280} are the maximum absorption of the fluorophore and the absorption at 280 nm, respectively, ϵ_{Label} the extinction coefficient of the fluorophore at its maximum, and ϵ_{NTS1} that of NTS1 at 280 nm ($56,840 \text{ M}^{-1}\text{cm}^{-1}$). CF is a correction factor to account for the contribution of the fluorophore at 280 nm which is determined from the excitation spectrum of the free fluorophore in solution as the percentage absorption at 280 nm compared to its maximum absorption.

→ Labelling efficiency was determined from the absorption spectra of the labelled protein samples, and discussion hereof as above has now been included in the methods section.

3.9 The authors always talk about receptor densities in the "physiological" range. Has the native NTS1 receptor density been determined? Was this density a mean density, and how does it vary depending on the cellular sub-compartment?

Reply: Cell membrane expression levels of the high-affinity neurotensin receptor NTS1 have been documented to range between 5,500-32,000 sites per cell in human endothelial cells (Schaeffer et al, JMB, 1995)²². Human endothelial cells have surface of $\sim 1000 \mu\text{m}^2$ (Jaffe, Human pathology, 1987)²⁴, giving an average NTS1 density of 5-30 copies/ μm^2 . Similar native densities have been found in other studies for NTS1 and other class A GPCRs (Gilbert et al 1986, Mazor et al 2002, Nakagawa et al 1984, Schonbrunn et al 1978, Shang et al 2015, Tennenberg et al 1988)^{19-21,25-27} – these studies are cited in our discussion section. Indeed, as the reviewer points out, these numbers represent a mean density, and do not exclude the possibility of higher local densities, which, according to our findings, could give rise to higher local dimer concentrations, as mentioned in the discussion. NTS1 has previously been proposed to be associated with “microdomains” on the plasma membrane (e.g. Heakal and Kester 2009; Heakal et al 2011)^{28,29}, but local densities have not been quantified. In fact, this questions from the reviewer hints at one of our key interpretations, that the ‘average’ NTS1 cellular density straddles a very sensitive region for concentration dependent dimerisation. Thus we may anticipate that any local variation, e.g. as the reviewer suggests by cellular sub-compartments, can be expected to have a significant effect on dimerisation.

→ We have clarified our discussion on the physiological receptor density of NTS1.

3.10 For class C GPCR, a rolling model for 7TM association has been reported, and a change in the dimer interface has been associated with receptor activity. These papers must be discussed (Xue et al., Nat Chem Biol 2015; Kim et al., PNAS 2017).

Reply: See section 1.3. Indeed, these, and other previously published studies have supported the notion of multiple dimerisation interfaces for GPCRs. However, these particular studies propose a change in dimerisation interface upon receptor activation, which is fundamentally different from the “rolling” interface that we see for NTS1 in the apo state by single-molecule FRET (for more detailed discussion, please see section 1.3). However, we agree that we need to better place these findings in the context of the current literature, and have expanded our discussion on this point.

→ We have altered our discussion of the rolling dimer model accordingly.

3.11 As based on the data shown in Fig 1c and d, the referee is not convinced of any significant difference between the diffusion coefficient of the dimers (acceptor diffusion) and monomers (donor diffusion).

Reply: As discussed under point 2.7(a), from the non-overlapping 95% confidence intervals for the diffusion constant of the spots in the acceptor ($[1.222, 1.317] \mu\text{m}^2\text{s}^{-1}$, $n = 1,167$ trajectories) and the donor channel ($[1.609, 1.647] \mu\text{m}^2\text{s}^{-1}$, $n = 17,853$ trajectories), we can conclude that the difference between them is significant. Indeed this was confirmed by a two-sided z-test ($z = -14.12$, $p < 10^{-5}$).

Our large number of observations (17,853 and 1,167 trajectories for donor and acceptor, respectively) enable us to establish the high degree of confidence in the difference in diffusion constant between the two species. With regard to interpretation of the MSD plots, it is worth noting that by their nature, we have much greater confidence and accuracy over short time

delays (and as is accepted practice, the fit is weighted accordingly). This is because i) displacement is calculated on a per trajectory basis for all combinations of delays available for that trajectory, thus there are many more combinations of short delay measurements than long delays even in long lasting trajectories. And this is compounded by the fact that the majority of measured trajectories are short as a result of photobleaching. And ii) The MSD plot describes measured *displacement* over time, rather than distance travelled. With displacement considered as the straight-line distance between points A and B, measurements are fundamentally limited by the time resolution of observation. Therefore, for any given rate of diffusion short delays (approaching the frame time) provide a closer approximation for distance travelled than long delays, as it is often tempting to interpret the MSD plot in this way. Consequently, we can expect to see a broadening of observed displacements (as reflected in the shaded area in our plots), as delays increase. This is because under a random walk, molecules may diffuse comparable *distances* yet be observed at very different *displacements* over the same time delay, and this effect increases as the delay increases. MSD is calculated for all available delays on an individual trajectory basis to generate the MSD vs. delay relationship for each trajectory, and the distribution of observed displacements at each delay across all trajectories is then used to form the MSD plot. As a consequence of i) and ii), it is expected that the standard deviation of observed displacements broadens as delay increases, even whilst retaining high confidence in the measured diffusion coefficient. Consequently, we consider overlap of these regions (shaded in grey in Fig. 1) between monomer and dimer at longer delays to not be unexpected. We consider the range of displacements at longer delays contributes additional valuable information in our plots, so have retained the detail, as it informs us that we have true Brownian diffusion with no net active movement or restriction in diffusion, which would manifest as net upward or downward (respectively) deflection in the MSD curve at longer delays.

→ The confidence intervals for the diffusion constants and the z-test statistics are now explicitly mentioned in the main text.

3.12 Page 7, second paragraph, please indicate the position of the labeling here so that the reader do have to look for it.

Reply:

→ The text has been adjusted as per the reviewer's recommendation.

Reviewer #4 (Remarks to the Author):

The authors studied dimerization of recombinant modified with fluorescent moieties or spin labels mutants of neurotensin receptor NTS1 in liposomes, using FRET and DEER. The authors conclude that NTS1 receptor dimerizes transiently, that the rate and extent of dimerization depend on receptor concentration in the membrane, and that multiple interfaces can be involved (rolling dimer interface model). They also found that at least one mode of dimerization is suppressed by the agonist. The authors hypothesize that dimerization can play a role in the regulation of receptor activity.

As far as the dynamics of receptor dimerization goes, the data with NTS1 support earlier findings with the M1 muscarinic (ref 28) and N-formyl peptide receptor (ref 29). Their inferences regarding the effect of dimerization on signaling are logical, but these are just inferences. In addition, several experimental, conceptual, and presentation issues are not addressed in the present version of the manuscript.

Experimental and conceptual issues:

4.1 The authors should present the data proving functionality of the purified labeled NTS1 constructs (at least their ability to bind neurotensin).

Reply: Please see point 2.3.

→ MST measurements demonstrating ligand and G protein binding have been added to the SI (Fig. S2 and S3) and are now mentioned in the main text of the manuscript.

4.2 The authors should discuss possible orientation of receptors in the lipid bilayer (if it is random, the calculations are off).

Reply: Please see point 3.3.

4.3 The authors should explain how their “rolling dimer interface” model is distinguished from the model of random receptor encounters without dimer formation, which would also involve different receptor surfaces. In particular, they should demonstrate that their data are not compatible with the latter model.

Reply: Please see point 2.1.

Our observations suggest that ~20% of monomer-monomer collisions result in productive dimer formation. This is (perhaps surprisingly) efficient and indicates the likely implication of more than one productive interface. Our data suggests that once formed, dimers are highly dynamic within the dimer lifetime (i.e. within single-molecule dimer trajectories) and this we interpret as the described “rolling dimer interface” model. To distinguish this from random receptor encounters that do not result in dimer formation, we have quantified the rate of false positive dimerisation detection (Supplementary Methods S2) visualised in an accompanying figure (Fig. S5) and added this to the SI. As discussed in detail under point 2.1a, we estimate a maximum false positive detection rate of 0.04% in our single-molecule FRET experiments, confirming that co-diffusing, non-interacting, species make a negligible contribution to dimerisation detections in our population of 1,167 measured trajectories. Most notably, this analysis demonstrates that non-dimer forming encounters would not persist on the time-scale of multiple frames and it is on this time-scale that we observe the dimer dynamics in individual dimer trajectories that we interpret as the described “rolling dimer interface” model.

4.4 The results of simulations should be clearly distinguished from the actual data, as simulations are based on certain assumptions, which may or may not be correct.

Reply:

→ The text and figure legends have been altered to make the distinction between in vitro and in silico experiments more clear.

4.5 The authors do not present any data on the effect of dimerization on signaling. Their inferences, however logical, should be clearly presented as speculation.

Reply:

→ The discussion has been altered to better reflect the speculative nature of our inferences on the functional implications of our findings.

Presentation issues:

4.6 In fact, not only G proteins were shown to couple to monomeric GPCRs in vitro, but also arrestins (J Mol Biol. 2010 Jun 11;399(3):501-11; J Biol Chem 2011 Jan 14;286(2):1420-8). Just like in case of G proteins (ref 53), these findings were supported by the crystal structure of the arrestin-receptor complex (Nature. 2015 Jul 30;523(7562):561-7).

Reply:

→ References of arrestin coupling to monomeric GPCRs have been added to the text.

4.7 The authors should clearly state that many GPCR crystal dimers are in a definitely non-physiological anti-parallel orientation. This suggests that one should interpret all dimers found in crystals with caution.

Reply: Indeed, reviewer #4 is correct in stating that many GPCR crystal “dimers” occur in antiparallel orientations, and are thus only due to crystal packing. Consequently, it is by no means certain that the parallel crystal “dimers” represent physiologically relevant, “real” dimers. At the very least they represent stable, energetically favourable interfaces given that the GPCRs in question were able to be crystallised in that configuration. Thus, in absence of other high-resolution information on GPCR dimerisation models we employed these parallel crystal dimers as just that, models of possible dimerisation interfaces. We agree that their nature is contentious, and have altered the text to alert the reader to that fact.

→ A paragraph stating these caveats has been added to the discussion.

4.8 Some editing is needed. There is a typo in ref 72; throughout the text, please correct verbs, as “data” is a plural noun, datum is a corresponding singular; etc.

Reply:

→ Text has been edited accordingly.

References:

1. Guo, W. *et al.* Dopamine D2 receptors form higher order oligomers at physiological expression levels. *EMBO J.* **27**, 2293–2304 (2008).
2. Xue, L. *et al.* Major ligand-induced rearrangement of the heptahelical domain interface in a GPCR dimer. *Nat. Chem. Biol.* **11**, 134–140 (2015).
3. Hu, J. *et al.* Novel structural and functional insights into M3 muscarinic receptor dimer/oligomer formation. *J. Biol. Chem.* **288**, 34777–34790 (2013).
4. Thompson, J. R., Cronin, B., Bayley, H. & Wallace, M. I. Rapid Assembly of a Multimeric Membrane Protein Pore. *Biophys. J.* **101**, 2679–2683 (2011).
5. Hern, J. A. *et al.* Formation and dissociation of M1 muscarinic receptor dimers seen by total internal reflection fluorescence imaging of single molecules. *Proc. Natl. Acad. Sci.* **107**, 2693–2698 (2010).
6. Altenbach, C., Kusnetzow, A. K., Ernst, O. P., Hofmann, K. P. & Hubbell, W. L. High-resolution distance mapping in rhodopsin reveals the pattern of helix movement due to activation. *Proc. Natl. Acad. Sci. U. S. A.* **105**, 7439–7444 (2008).
7. White, J. F. *et al.* Structure of the agonist-bound neurotensin receptor. *Nature* **490**, 508–513 (2012).
8. Egloff, P. *et al.* Structure of signaling-competent neurotensin receptor 1 obtained by directed evolution in *Escherichia coli*. *Proc. Natl. Acad. Sci. U. S. A.* **111**, E655–E662 (2014).
9. Krumm, B. E., White, J. F., Shah, P. & Grisshammer, R. Structural prerequisites for G-protein activation by the neurotensin receptor. *Nat. Commun.* **6**, 7895 (2015).
10. Krumm, B. E. *et al.* Structure and dynamics of a constitutively active neurotensin receptor. *Sci. Rep.* **6**, 38564 (2016).
11. Casciari, D., Dell’Orco, D. & Fanelli, F. Homodimerization of neurotensin 1 receptor involves helices 1, 2, and 4: insights from quaternary structure predictions and dimerization free energy estimations. *J. Chem. Inf. Model.* **48**, 1669–1678 (2008).
12. Ramadurai, S. *et al.* Lateral diffusion of membrane proteins. *J. Am. Chem. Soc.* **131**, 12650–12656 (2009).
13. Harding, P. J. *et al.* Constitutive dimerization of the G-protein coupled receptor, neurotensin receptor 1, reconstituted into phospholipid bilayers. *Biophys. J.* **96**, 964–973 (2009).
14. Bouvier, M. & Hébert, T. E. CrossTalk proposal: Weighing the evidence for Class A GPCR dimers, the evidence favours dimers. *J. Physiol.* **592**, 2439–41 (2014).
15. Gurevich, V. V & Gurevich, E. V. How and why do GPCRs dimerize? *Trends Pharmacol. Sci.* **29**, 234–240 (2008).
16. Kasai, R. S. *et al.* Full characterization of GPCR monomer-dimer dynamic equilibrium by single molecule imaging. *J. Cell Biol.* **192**, 463–480 (2011).
17. Calebiro, D. *et al.* Single-molecule analysis of fluorescently labeled G-protein-coupled receptors reveals complexes with distinct dynamics and organization. *Proc. Natl. Acad. Sci.* **110**, 743–748 (2013).
18. Bolivar, J. H. *et al.* Interaction of lipids with the neurotensin receptor 1. *Biochim. Biophys. Acta - Biomembr.* **1858**, 1278–1287 (2016).
19. Gilbert, J. A., Moses, C. J., Pfenning, M. A. & Richelson, E. Neurotensin and its analogs—correlation of specific binding with stimulation of cyclic GMP formation in neuroblastoma clone N1E-115. *Biochem. Pharmacol.* **35**, 391–397 (1986).
20. Mazor, O. *et al.* Europium-labeled epidermal growth factor and neurotensin: novel probes for receptor-binding studies. *Anal. Biochem.* **301**, 75–81 (2002).
21. Nakagawa, Y., Higashida, H. & Miki, N. A single class of neurotensin receptors with high affinity in neuroblastoma X glioma NG108-15 hybrid cells that mediate facilitation of synaptic transmission. *J. Neurosci.* **4**, 1653–1661 (1984).
22. Schaeffer, P. *et al.* Human umbilical vein endothelial cells express high affinity neurotensin receptors coupled to intracellular calcium release. *J. Biol. Chem.* **270**, 3409–3413 (1995).
23. Pluhackova, K., Gahbauer, S., Kranz, F., Wassenaar, T. A. & Böckmann, R. A. Dynamic Cholesterol-Conditioned Dimerization of the G Protein Coupled Chemokine Receptor Type 4. *PLoS Comput. Biol.* **12**, 1–25 (2016).
24. Jaffe, E. A. Cell biology of endothelial cells. *Hum. Pathol.* **18**, 234–239 (1987).
25. Schonbrunn, A. & Tashjian, A. H. Characterization of functional receptors for somatostatin in rat pituitary cells in culture. *J. Biol. Chem.* **253**, 6473–6483 (1978).

26. Shang, Y. & Filizola, M. Opioid receptors: Structural and mechanistic insights into pharmacology and signaling. *Eur. J. Pharmacol.* 1–8 (2015). doi:10.1016/j.ejphar.2015.05.012
27. Tennenberg, S. D., Zemlan, F. P. & Solomkin, J. S. Characterization of N-formyl-methionyl-leucyl-phenylalanine receptors on human neutrophils. Effects of isolation and temperature on receptor expression and functional activity. *J. Immunol.* **141**, 3937–3944 (1988).
28. Heikal, Y. & Kester, M. Nanoliposomal short-chain ceramide inhibits agonist-dependent translocation of neurotensin receptor 1 to structured membrane microdomains in breast cancer cells. *Mol. Cancer Res.* **7**, 724–734 (2009).
29. Heikal, Y. *et al.* Neurotensin receptor-1 inducible palmitoylation is required for efficient receptor-mediated mitogenic-signaling within structured membrane microdomains. *Cancer Biol. Ther.* **12**, 427–435 (2011).

REVIEWERS' COMMENTS:

Reviewer #2 (Remarks to the Author):

I have gone through the point by point response . I think the authors have satisfactorily addressed all the points that I raised.

Reviewer #3 (Remarks to the Author):

The authors adequately respond to most of my remarks. However, in my opinion, if the authors' proposal is correct –i.e. that receptors can associate through different interfaces - I remain convinced that such a process is very unlikely to be of any strong functional consequence, or at least that the functional consequence is unlikely the same depending on the interface involved in the association. Indeed, such receptor association is likely to have functional consequences if allosteric interaction can occur between the protomers, and it is obvious that such interaction are not expected to be the same depending on the interaction interface. This point must be clearly discussed in the manuscript.

Specific comments:

Discussion, L330: I m not satisfied by this sentence as the authors mainly examined the dynamics and mode of interaction of NTS1R, but do not examine how such process "modulates GPCR function".

Discussion, L355: one cannot exclude the possibility that two receptors can interact even when differently oriented in the plasma membrane, as suggested by X ray data revealing such inappropriate GPCR dimers.

Reviewer #4 (Remarks to the Author):

The authors presented a comprehensive analysis of recombinant modified with fluorescent moieties or spin labels mutants of neurotensin receptor NTS1 in liposomes, using FRET and DEER. The authors conclude that NTS1 receptor dimerizes transiently, that both the rate and extent of dimerization depend on receptor concentration in the membrane, and that multiple interfaces are likely involved (rolling dimer interface model). They also found that at least one mode of dimerization, involving TM6, which moves upon GPCR activation, is suppressed by the agonist. The manuscript became much stronger after revision, but that actually emphasized the issue of novelty (or, rather, lack thereof). The dynamics of NTS1 receptor dimerization described here are consistent with earlier findings with the M1 muscarinic (ref 29) and N-formyl peptide receptor (ref 30). The data with NTS1 receptor reported here do not move the field beyond those earlier findings. The term "rolling interface" is novel, but the phenomenon was described earlier. The authors hypothesize that dimerization can play a role in the regulation of receptor activity, but this is just a speculation. Presented structural models are just that, models, with uncertain predictive potential.

Thus, while the specific criticisms were adequately addressed, overall this manuscript is not a serious breakthrough one expects from Nature Communications. It appears better suited for a more specialized journal.

Reviewer #2 (Remarks to the Author):

I have gone through the point by point response . I think the authors have satisfactorily addressed all the points that I raised.

Reviewer #3 (Remarks to the Author):

The authors adequately respond to most of my remarks. However, in my opinion, if the authors' proposal is correct –i.e. that receptors can associate through different interfaces - I remain convinced that such a process is very unlikely to be of any strong functional consequence, or at least that the functional consequence is unlikely the same depending on the interface involved in the association. Indeed, such receptor association is likely to have functional consequences if allosteric interaction can occur between the protomers, and it is obvious that such interaction are not expected to be the same depending on the interaction interface. This point must be clearly discussed in the manuscript.

Reply: We agree with the reviewer that the functional consequence of dimerisation may depend on the interface involved in the association. Indeed, as we argue with Figure 6, not all dimer interfaces may be able to support signalling through G proteins and that, for example, an interface made by TM5-6 may limit basal G protein coupling activity of the receptor. As before, we would argue that the possibility of different dimer interfaces may in fact be important for GPCRs which show great functional plasticity, signalling through a wide array of pathways. Particular dimer interfaces may be more energetically favourable in certain cellular environments, for example by interactions with different lipid environments¹ or other cellular components, which may alter the functional consequence of dimerisation depending on the interface that is favoured. We have devoted an entire paragraph of the discussion to these points, but to aid the reader have now also specifically stated in the beginning of said paragraph that “the functional consequence of dimerisation may depend on the interface involved in the association”.

Specific comments:

Discussion, L330: I m not satisfied by this sentence as the authors mainly examined the dynamics and mode of interaction of NTS1R, but do not examine how such process "modulates GPCR function".

Reply: We have changed the sentence to reflect our focus on NTS1, by replacing “GPCRs” with “NTS1”.

Discussion, L355: one cannot exclude the possibility that two receptors can interact even when differently oriented in the plasma membrane, as suggested by X ray data revealing such inappropriate GPCR dimers.

Reply: Indeed, we cannot entirely exclude this possibility, but as previously argued, our observations support the conclusion that the dimer species we observe must originate from parallel dimers. Specifically,

“The observed TM4-4 distance of ~5.0 nm for the low FRET state and shorter (~1.5 nm) for the high-FRET state are too short to both originate from anti-parallel dimers, for which we would expect a minimal distance of ~4-7 nm. Thus, neither the high-FRET state observed in smFRET, nor the short distances observed by DEER, could originate from an anti-parallel dimer. Furthermore, the smFRET threshold crossing analysis (presented in Fig. 4) showed that individual dimers sample both the low- and high-FRET states, i.e. show interconversion between these states. Both these states must thus originate from a parallel dimer to be able to interconvert, validating our conclusion that the physiologically relevant dimer samples multiple interfaces.”

However, to clarify this point for the reader, we have now included our extended arguments as Supplementary Note 7, and point the reader to this Note in the text.

Reviewer #4 (Remarks to the Author):

The authors presented a comprehensive analysis of recombinant modified with fluorescent moieties or spin labels mutants of neurotensin receptor NTS1 in liposomes, using FRET and DEER. The authors conclude that NTS1 receptor dimerizes transiently, that both the rate and extent of dimerization depend on receptor concentration in the membrane, and that multiple interfaces are likely involved (rolling dimer interface model). They also found that at least one mode of dimerization, involving TM6, which moves upon GPCR activation, is suppressed by the agonist. The manuscript became much stronger after revision, but that actually emphasized the issue of novelty (or, rather, lack thereof). The dynamics of NTS1 receptor dimerization described here are consistent with earlier findings with the M1 muscarinic (ref 29) and N-formyl peptide receptor (ref 30). The data with NTS1 receptor reported here do not move the field beyond those earlier findings. The term “rolling interface” is novel, but the phenomenon was described earlier. The authors hypothesize that dimerization can play a role in the regulation of receptor activity, but this is just a speculation. Presented structural models are just that, models, with uncertain predictive potential.

Thus, while the specific criticisms were adequately addressed, overall this manuscript is not a serious breakthrough one expects from Nature Communications. It appears better suited for a more specialized journal.

Reply: We thank the reviewer for their positive remarks and previous comments and suggestions which we agree improved the manuscript. However, we disagree with the reviewer and believe that our results have sufficient merit to be of interest to a more general audience, demonstrating as they do some novel information about the dynamics of dimer interactions, as well as a significant extension of the methodology.

Firstly, while indeed our data on NTS1 dimerisation dynamics are consistent with previous results, it has to be noted that earlier studies on GPCR dimerisation have relied on fluorescent co-localisation, while here we employ FRET. As argued in the discussion section, the use of FRET greatly enhances the spatial resolution (<10 nm) compared to diffraction-limited methods (>200 nm). FRET thus presents an advantage in reliably determining protein-protein interactions over diffraction-limited co-localisation techniques which may interpret protomer crowding as oligomerisation. The use of FRET thus improves the accuracy of bona fide dimer detection. While a more recently published study reported on GPCR dimerisation using super-resolution imaging to improve spatial resolution of co-localisation², the reduced temporal resolution precluded kinetic studies of dimer formation. Thus, the combination greater spatial and temporal resolution of the work presented here allows us to confirm previously reported results for studies on other GPCRs and lend more credibility to the notion of transient GPCR dimerisation due to improved methodology in a still controversial field.

Additionally, here, we characterised intrinsic receptor-receptor interactions, unperturbed by additional cellular interactions (accessory proteins, cytoskeleton etc.), suggesting that transient dimerisation is an inherent property of the receptor, which is a first demonstration of this kind. While we cannot directly extrapolate from NTS1 to all GPCRs, the good agreement with previous studies on transient dimerisation suggest that dimerisation may be a truly intrinsic property of other GPCRs as well, ruling out artefactual observation (e.g. crowding) or biological mechanisms of local corralling that cannot be discerned by previous co-localisation studies in cells, even at the single molecule level.

Combined with the observation that oligomerisation is strongly influenced by receptor density, we feel that these findings rationalise the wide disparity in views about oligomerisation in the GPCR literature,

where different extents of oligomerisation have been reported, depending on the (methodologically required) receptor density used, and will enlighten much of the current literature.

Further, our use of single-molecule FRET, not only affords increased confidence of bona fide receptor-receptor interaction; it also has allowed us to make dynamic measurements on TM-TM separation distance on the timescale of ~30 ms in individual dimer complexes – this is not accessible by single-molecule tracking alone. This, in conjunction with the ensemble techniques we employ not only provides evidence for multiple dimerisation interfaces, but provides evidence for the first time that suggests these do not exist as distinctly separate species but are instead able to *interconvert* dynamically within a single dimer receptor pairing.

While the possibility of multiple dimerisation interfaces has been previously proposed, as we summarise in our manuscript, in our view, with the data presented here, we extend the model from the possibility of the formation of dimers in different configurations that *do not interconvert directly*, to the formation of a metastable dimer, which samples multiple interfaces with varying stability on a per molecule basis. Or in other words: our “rolling dimer” interface differs in that it is not different dimers that are present in different configurations (i.e. having different dimer interfaces), but discreet dimers that sample many states within the dimer lifetime.

Furthermore, previous studies proposed a change in dimerisation interface upon receptor activation, which is fundamentally different from the “rolling” interface that we see for NTS1 in the apo state by single-molecule FRET. To the best of our knowledge this is a new nuance in the discussion of GPCR dimer interfaces that could have significant implications for the understanding of both the mechanisms and functional consequences of GPCR dimerisation, and may help researchers interpret their own findings in new ways.

Indeed, the present work does not attempt to answer the functional role of dimerisation, nor does it provide conclusive models of NTS1 dimers, which we feel would go beyond the scope of the current study. However, we feel that the work does provide a lot of scope for future research in these areas.

References

1. Pluhackova, K., Gahbauer, S., Kranz, F., Wassenaar, T. A. & Böckmann, R. A. Dynamic Cholesterol-Conditioned Dimerization of the G Protein Coupled Chemokine Receptor Type 4. *PLoS Comput. Biol.* **12**, 1–25 (2016).
2. Jonas, K. C., Fanelli, F., Huhtaniemi, I. T. & Hanyaloglu, A. C. Single molecule analysis of functionally asymmetric G protein-coupled receptor (GPCR) oligomers reveals diverse spatial and structural assemblies. *J. Biol. Chem.* **290**, 3875–3892 (2015).